# Measurement Report: Insights into the chemical composition and origin of molecular clusters and potential precursor molecules present in the free troposphere over the Southern Indian Ocean: observations from the Maïdo observatory (2150 m a.s.l., Reunion Island)

Romain Salignat[1], Matti Rissanen[2,3], Siddharth Iyer[2,3], Jean-Luc Baray[1], Pierre Tulet[4], Jean-Marc Metzger[5], Jérôme Brioude[6], Karine Sellegri[1], Clémence Rose[1]

[1]Université Clermont Auvergne, CNRS, Laboratoire de Météorologie Physique (LaMP), F-63000 Clermont-Ferrand, France.
[2]Aerosol Physics Laboratory, Faculty of Engineering and Natural Sciences, University of Tampere, 33720 Tampere, Finland.
[3]Chemistry Department, Molecular Research Unit, University of Helsinki, 00014 Helsinki, Finland.
[4]Laboratoire d'Aérologie (LAERO), Université Paul Sabatier, CNRS, IRD, Toulouse, France.
[5]Observatoire des Sciences de l'Univers de La Reunion, UAR 3365 (CNRS, Université de La Reunion, Météo-France), 97744 Saint-Denis, Reunion, France.
[6] Laboratoire de l'Atmosphère et des Cyclones (LACy), UMR 8105, Université de la Reunion, CNRS, Saint–Denis, France.

*Correspondence to*: c.rose@opgc.fr

## Abstract

New particle formation (NPF) in the free troposphere (FT) is thought to be a significant source of particles over the oceans. The entrainment of particles initially formed in the marine FT is further suspected to be a major contributor to cloud condensation nuclei (CCN) number concentrations in the marine boundary layer (BL). Yet, little is known about the process, and more broadly about the composition of the marine FT, which remains poorly explored due to access difficulties. Here we report measurements performed in April 2018 at the Maïdo observatory with a nitrate based chemical ionisation atmospheric pressure interface time-of-flight mass spectrometer, which have allowed the first molecular-level characterisation of the remote marine FT composition. A number of molecules and clusters were identified and classified into 9 groups according to their chemical composition; among the identified species, the groups containing methanesulfonic acid (MSA) and C2 amines show signals that are on average significantly higher when the site is under conditions representative of the marine FT (compared to the BL). The correlation analysis revealed apparent connections between the signals of the identified compounds and several variables concurrently measured at the site (under FT conditions) or related to air mass history, suggesting that oxalic acid, malonic acid and observed C2 amines could be of terrestrial origin, with, in addition, a possible marine source for oxalic acid and amines, while iodic acid, sulfur species and maleic acid have a dominant marine origin. Identification of FT conditions at the site was based on the analysis of the standard deviation of the wind direction; this parameter, which can easily be derived from continuous measurements at the site, is shown in the first part of the study to be a relevant tracer when compared to predictions from the Meso-NH atmospheric model. Similar to other high altitude sites, FT conditions are mainly encountered at night at Maïdo and therefore the link to NPF could not be established, and further research is needed to assess the composition of precursors to nanoparticle formation in the marine FT.

## 1. Introduction

Aerosol particles are a core component of the climate system, in particular due to their role in cloud formation and the subsequent impact they have on their properties, including their albedo (Twomey et al., 1977) and life-time (Albrecht, 1989). Although the related atmospheric processes have been widely studied, the associated effects on the Earth's radiation balance (known as the indirect effects) still represent one of the largest sources of uncertainty in climate projections (IPCC, 2013). Models predict that around half of the present-day cloud forming particles may originate from new particle formation (NPF) at the global scale (Merikanto et al., 2009; Gordon et al., 2017), i.e. from gas-to-particle conversion processes. NPF has been

extensively studied during the last decades, and reported to occur in almost all known environments (Kerminen et al., 2018). Although there are still gaps in our knowledge, the development of mass spectrometry techniques achieved at the same time has improved our understanding of the mechanisms by giving access to the chemical composition of the molecular clusters and their precursors (Junninen et al., 2010; Jokinen et al., 2012). Besides sulfuric acid, which is commonly accepted as a key driver of NPF, the nucleating ability of other compounds has been demonstrated, both from observations in the real atmosphere and experiments in simulation chambers. The role of ammonia (Kirkby et al., 2011) and amines (Almeida et al., 2013) in stabilizing sulfuric acid clusters has in particular been highlighted, and the potential of organic compounds of biogenic origin to also contribute (Schobesberger et al., 2013; Riccobono et al., 2014; Lehtipalo et al., 2018) and even nucleate on their own (Kirkby et al., 2016; Rose et al., 2018) has been reported. In the marine environment, although observations are limited, the role of iodic acid has been more specifically identified in coastal areas (Sipilä et al., 2016; Baccarini et al., 2020; Beck et al., 2021; Peltola et al., 2023). However, most observations in the real atmosphere have been made in the planetary boundary layer (BL), i.e. in the lower part of the troposphere that is directly influenced by the earth's surface (Stull, 1988). Observations are in contrast more scarce in the free troposphere (FT), i.e. at higher tropospheric altitudes above the BL, mainly due to the difficulty of sampling this remote region of the atmosphere. Yet, several studies report the occurrence of NPF, or at least its early stages, in the low FT, or at the interface between the BL and the FT (Bianchi et al., 2016; Boulon et al., 2011; Rose et al., 2015a,b; Rose et al., 2017), where the conditions, including higher radiation, lower temperature and reduced condensation sink for the nucleating species, could favour the occurrence of the process (Rose et al., 2015a; Sellegri et al., 2019).

In particular, NPF could be promoted in the marine FT (MFT) compared to the marine BL (MBL), and the entrainment of growing particles formed in the MFT is further suspected to be a major contributor to MBL cloud condensation nuclei (CCN) number concentration (Clarke et al., 2013; Quinn et al., 2017; Williamson et al., 2019; McCoy et al., 2021). Over the remote ocean, the available observations suggest that oxidation products of dimethylsulfide (DMS) emitted by phytoplankton, and in particular sulfuric acid ($H_2SO_4$), may play a central role in MFT NPF (Clarke et al., 1998b; Weber et al., 2001; Quinn et al., 2017; McCoy et al., 2021). There is, however, no direct evidence of the implication of sulfuric acid in the process since these MFT NPF events have not been the subject of any detailed chemical characterization. Recent measurements performed in the South Pacific Ocean (Peltola et al., 2022; 2023) have enabled an important step forward in documenting NPF in the remote MBL. Long term observations conducted at the Baring Head Global Atmosphere Watch (GAW) station in New Zealand have revealed that MBL NPF occurs frequently over the South Pacific Ocean, and contributes significantly to the total aerosol number concentration, in particular during spring time (Peltola et al., 2022). Mass spectrometry analysis indicate that sulphur species resulting from DMS oxidation likely contribute to the composition of newly formed charged clusters during daytime, together with iodine oxides (Peltola et al., 2023). These observations are consistant with the results reported from cruises in the Southern Ocean (Baccarini et al., 2021; Brean et al., 2021), although these measurements were mainly performed in coastal and/or ice-influenced air masses. The relevance of these results at higher altitudes, and in particular in the FT, where conditions are different from those at the surface, is however not implicit. For example, it was recently shown by Beck et al. (2022) that cluster ions of different chemical composition are found in each of the lowest atmospheric layers above the boreal forest, reflecting the presence of precursors of varying nature, in connection with air mass origin and time of the day. We expect that some differences also exist over the remote ocean, and the characterization of MFT NPF therefore remains an open field.

The current experimental knowledge of MFT NPF is based on airborne measurements (e.g. Clarke et al., 1998b; Weber et al., 2001; Rose et al., 2015a). Besides the fact that it is challenging to operate the mass spectrometers used for the chemical characterisation of the clusters on airborne platforms, such measurements can only provide snapshots of the FT. Although they are not continuously in the FT due to the combination of boundary layer dynamics and topographic effects (Collaud Coen et al., 2018), mountain observatories offer the possibility of conducting measurements over longer periods, and also to deploy more extensive instrumental set-ups. With the combined use of a proton transfer reaction time-of-flight mass spectrometer (PTR-MS; Hansel et al., 1995) and a nitrate chemical ionization atmospheric-pressure-interface time-of-flight mass

spectrometer (CI-APi-TOF; Jokinen et al., 2012), Scholz et al. (2023) have recently reported the presence of DMS and most of its known oxidation products at the GAW Andean high-altitude station of Chacaltaya (5240 m a.s.l., Bolivia) after they have been transported in the FT from the Pacific Ocean. This station is however located 330 km inland and therefore the concentrations of marine compounds that are measured there are low (due to dilution and deposition losses during transport), and a priori insufficient to impact NPF at the site. Also, due to its continental location, the site is subject to the influence of other sources (biogenic, anthropogenic and volcanic) which limit the possibility of studying compounds of marine origin and their possible involvement in NPF.

There are not many measurement stations that are at a sufficiently high altitude and representative of the surrounding ocean around the world. Here we report measurements performed at the GAW high-altitude observatory of Maïdo (2150 m a.s.l.; Baray et al., 2013) on Reunion island, in the southern Indian Ocean. Due to the small size of the island (2512 km²) and its geographical location, the sampling frequency of MFT air masses from Maïdo is much higher than from Chacaltaya. In addition, the southern Indian Ocean is one of the few regions on Earth that is only slightly impacted by human activity and is characterized by conditions often close to those encountered in the pre-industrial era (Hamilton et al., 2014), making it an ideal region for the study of marine aerosol production and sources. The first part of this study is dedicated to investigating the suitability of existing methods to detect in a simple way, from the continuous measurements performed at the site, the periods during which the station is in the FT (Sect. 3.1). An analysis of the ability of the CAT (Computing Advection-interpolation of atmospheric parameters and Trajectory tool; Baray et al., 2020) trajectory model to assess in addition the positioning of air masses in the BL or in the FT during the hours preceding their arrival at the site is also presented (Sect. 3.2). The second part of this work provides insights into the chemical composition of molecular clusters and potential precursor molecules detected in the MFT (Sect. 4). Since, as for all high altitude sites, FT conditions are mainly encountered at night at Maïdo and our data set was of limited length, the involvement of the identified compounds in NPF, which takes place preferentially during the day, could not be addressed. However, despite this limitation, the observations reported here make an important contribution to the documentation of the MFT chemistry and physics, and indirectly to the understanding of the processes that take place in this specific region of the atmosphere.

## 2. Materials and methods

### 2.1 Observations

#### 2.1.1 Atmospheric dynamics in Reunion

The measurements used in the present work were performed at the high-altitude observatory of Maïdo, which is located on Reunion Island, about 15 km inland from the west coast (21.080◦ S, 55.383◦ E; 2150 m a.s.l.). Reunion Island is located in the Indian Ocean, in the descending part of the southern Hadley cell, and is therefore subject to a southeasterly trade wind flow in the low layers (induced by the Hadley cell and reinforced by the Walker circulation) (Baldy et al., 1996). Above this low-level flow are the westerly winds that constitute the Hadley-Walker return circulation, the intensity of which vary according to the position, direction and intensity of the subtropical jet stream. As described by Lesouëf et al. (2011), the atmospheric circulation at the local scale is to a large extent conditioned by the action of the island's relief and the radiation on the synoptic flow. Indeed, Reunion has a complex topography organized around two mountainous zones (with the highest summit, the Piton des neiges, peaking at 3069 m) which constitute an obstacle which forces the approaching trade winds to split into two branches (Duflot et al., 2019; El Gdachi, submitted). Reunion is in addition impacted by thermal breezes (sea and land breezes, slope breezes) that follow a daily cycle set by the sun and modulate the synoptic flow around the island. Its complex topography, together with the resulting local circulation as well as the diversity of vegetation cover, are as many factors that impact the development of the BL over Reunion.

The pioneering case study of Lesouëf et al. (2013) has allowed, based on a model-measurement synergy, to get particular insight into the development of the BL on the slopes of Maïdo. Detailed characterisation of the early morning (07:00 – 08:00 LT) circulation over the mountain's western slope reveals the presence of two atmospheric layers on the chosen day (November 26$^{th}$, 2008). Moist marine air is advected in the lower layer (∼1600 m thick) as a result of wake vortices in the lee of the island, while the upper layer corresponds to FT air driven by the easterly trade winds. In the following hours, the model indicates that the circulation near the surface is dominated by upward flow which brings humid air to the station's altitude. As a result, the observatory lies under the influence of low-level air between mid-morning and early evening, before trade winds evacuate humid air and replaces it with FT air once upward transport has ceased. Studies conducted since the work of Lesouëf et al. (2013) have further highlighted the complexity of the regime that actually impacts Maïdo during the day (Duflot et al., 2019; Leriche et al., 2023; El Gadchi et al., submitted). The station lies at the confluence of thermal breezes from the west and the easterly trade winds, and it is the interaction between the trade winds and the island's relief which modulates the influence of these two flows on the site.

### 2.1.2 Measurement site and instrumentation

The data sets obtained in the framework of two campaigns conducted at Maïdo were used in particular in the present work. Data collected during the BIO-MAIDO campaign (Leriche et al., 2023; https://anr.fr/Project-ANR-18-CE01-0013, last access: April 7, 2023) between March 14 and April 8 2019 were first used in synergy with the results of the high spatial resolution simulations conducted with the mesoscale atmospheric model Meso-NH (Lac et al., 2018) for the same period, in order to identify a tracer of FT conditions at the site (Sect. 3.1). The chemical composition of molecular clusters and their precursors were in a second step investigated using measurements performed with an extended instrumental setup deployed at the site in 2018 (from March to May, focus on the period between April 11 and 17 in the present study), during the OCTAVE (Oxygenated organic Compounds in the Tropical Atmosphere: variability and atmosphere–biosphere Exchanges; http://octave.aeronomie.be, last access: April 7, 2023) campaign (Sect. 4). The overall instrumental setup used in this work is described below, and the availability of each instrument during the two campaigns is summarized in Table 1. A more global description of the facility can be found in Baray et al. (2013).

The particle number size distribution between 10 and 600 nm is continuously monitored at the station (with a time resolution of 7 min) with a custom-built differential mobility particle sizer (DMPS) comprising an Ni-63 bipolar charger at 95 MBq, a TSI-type differential mobility analyzer (DMA) operating in a closed loop and a condensation particle counter (CPC, TSI model 3010). The DMPS was operated behind a whole air inlet which is characterised by a higher size cut-off of 25 µm for an average wind speed of 4 m s$^{-1}$. The air is dried before entering the instrument, so that the relative humidity of the sample is kept below 40%, in line with the recommendations of Wiedensohler et al. (2012). DMPS measurements were used in the calculation of the condensation sink (CS), which represents the loss rate of vapours on pre-existing aerosol particles (Kulmala et al., 2012), and the possibility of their use as a tracer of FT conditions has also been studied (see Sect. 3.1). Meteorological parameters, which are also recorded on an ongoing basis (with a time resolution of 3 s), were used as well, including global radiation (SPN1, DeltaT Devices Ltd., resolution 0.6 Wm$^{-2}$), temperature, relative humidity (RH), wind speed and direction (Vaisala Weather Transmitter WXT510).

As previously introduced in Rose et al. (2021), a nitrate CI-APi-TOF (Jokinen et al., 2012) was deployed during OCTAVE for the chemical characterization of neutral molecules and clusters. The CI-APi-TOF is the combination of an atmospheric pressure interface time of flight mass spectrometer (APi-TOF; Junninen et al., 2010) and a chemical ionisation (CI) inlet, in which nitric acid is charged (with a soft X-ray source) and deployed to ionize gas molecules, either by proton transfer or cluster formation. The total sampling flow was 10 L min$^{-1}$, of which 0.8 L min$^{-1}$ actually entered the instrument. The data we report are averaged over 3 min.

The observed signals were normalized by the sum of the reagent ion (i.e. $NO_3^-$, $HNO_3 \cdot NO_3^-$, $(HNO_3)_2 \cdot NO_3^-$) signals. A calibration coefficient of $1.7 \times 10^{10}$ molec.cm$^{-3}$ (including sampling loss correction) was determined on-site at the end of the campaign for $H_2SO_4$ following the approach of Kürten et al. (2012). Similar to Baccarini et al. (2021), the same coefficient was applied to estimate methanesulfonic acid (MSA) and iodic acid ($HIO_3$) concentrations, assuming that ionization proceeds

at the kinetic limit for these species which have a lower proton affinity than nitric acid (Shen et al., 2022; Finkenzeller et al., 2023). For amines, in contrast, in the same way as Brean et al. (2021), we only report ion signals since we did not perform specific in situ calibration for these compounds. In Fig. 6, which presents the average spectrum obtained in FT conditions and the difference of the average spectra obtained in the FT and in the BL, the signals of the amines have been normalised by the sum of the signals of all the reagent ions, as the signals of all other compounds shown in these figures. In Figs. 8 and 9,

however, where the time series of the amine signals are shown independently, the signals were normalised to the ionized nitric acid trimer ($(HNO_3)_2 \cdot NO_3^-$) signal only, according to the recommendations of Simon et al. (2016) that were also followed by Brean et al. (2021).

### 2.2 Models

#### 2.2.1    Meso-NH

Meso-NH is a non-hydrostatic mesoscale atmospheric model which has been co-developed by Laboratoire d'Aérologie (LAERO, UMR 5560 UPS/CNRS/IRD) and Centre National de Recherches Météorologiques (CNRM, UMR 3589 Météo-France/CNRS; Lac et al.,  2018; http://mesonh.aero.obs-mip.fr/, last access: April 7, 2023). The ability of Meso-NH (run with a typical horizontal spatial resolution of 500 m) to reproduce the main details of the atmospheric flows over Reunion was first

demonstrated by Lesouëf et al. (2011; 2013), and the model has been used again more recently to characterize the air masses over the island by Duflot et al. (2019) and El Gdachi et al. (submitted). The model configuration used for the simulations performed in the framework of BIO-MAIDO is described in Rocco et al. (2022). Briefly, three nested domains were used (Fig. S1), including two large domains covering the entire island with horizontal spatial resolutions of 2 km and 500 m, respectively, and a smaller domain (~10km wide) centered around the site with a particularly high spatial resolution of 100 m to enhance

turbulence (Large Eddy Simulation), clouds and local circulations such as slopes breezes. The same vertical grid comprising 72 stretched levels was used for the three domains. The initial and boundary meteorological conditions were supplied by the AROME operational high-resolution analysis with a temporal resolution of 6 h and a horizontal spatial resolution of 2.5 km. Specific interest was paid to the thickness of the BL simulated by the model. The BL thickness retrieved  at the location of the station in the innermost domain was considered first, to determine whether the model predicts the site to be in the FT or in the

BL and further use this information to determine a tracer of FT conditions from continuous measurements at the site. The BL thickness simulated on the path of the air masses arriving at the station in the larger domains was used in a second step to study air mass history. Determination of the BL thickness in the model is based on the bulk Richardson method, initially proposed by Vogelezang and Holtslag (1996). Among the methods available in the literature, the bulk Richardson method (with several approximations applied to the original algorithm) was reported by Seidel et al. (2012) to be the most suitable for characterizing

BL heights (BLH) from large radiosonde, reanalysis, and climate model data sets. The actual BLH  was derived by summing the simulated BL thickness and the ground elevation of the corresponding grid point in the model. The elevation of the grid point which contains the Maïdo observatory in the innermost domain is 2146 m a.s.l., reflecting the high spatial resolution and resulting remarkable accuracy of the topography in this domain.

#### 2.2.2    CAT and Meso-CAT trajectory models

The CAT (Computing Advection-interpolation of atmospheric parameters and Trajectory tool, Baray et al., 2020) model is a recent evolution of the 3D kinematic trajectory code LACYTRAJ (Clain et al., 2010), which computes air mass back-trajectories using wind fields from the ECMWF ERA-5 reanalyses (horizontal spatial resolution of 0.25°) and a topography

matrix with a grid resolution of about 10 km (Bezděk and Sebera, 2013) . This model has already been used for a statistical analysis of the air masses arriving at the puy de Dôme station (France, 1465 m a.s.l.; Baray et al., 2020). In the present work, 72-hour backward trajectories were calculated every hour from April 11, 2018 to April 17, 2018 with a resolution of 15 min. More specifically, trajectory sets comprising 125 trajectories in a 3D starting domain centered around the station (5×5×5 trajectories in a domain of horizontal and vertical dimensions of 20 km and 200 m, respectively) were considered. The history of air masses arriving at Maïdo in the FT was more particularly investigated in relation to their positioning in the BL or in the FT along their path. On each back-trajectory point, the altitude of air masses was deduced from the pressure fields and compared to the BLH calculated from the BL thickness retreived by the ECMWF ERA-5 reanalyses (calculated with the bulk Richardson method, similar to Meso-NH) and the topography matrix from Bezděk and Sebera (2013).

We also performed a sensitivity study with the Meso-CAT model (Rocco et al., 2022). Meso-CAT is an enhanced version of CAT involving higher resolution atmospheric parameters and topography (horizontal spatial resolution of 500 m) from Meso-NH, which are expected to be more relevant to the steep relief of Reunion. Twelve hour back-trajectories were calculated with Meso-CAT every 15 min with a temporal resolution of 5 minutes in the framework of the BIO-MAIDO campaign, and compared to the results provided by CAT (Sect. 3.2). More specifically, trajectory sets comprising 324 (9×9×4) trajectories in a starting domain (of horizontal and vertical dimensions of 500 m and 75 m, respectively) centered around the station, were considered in Meso-CAT. The purpose of this sensitivity study was in particular to evaluate the ability of CAT (which has a lower spatial resolution compared to Meso-CAT but can be run easily using ECMWF ERA-5 reanalyses) to assess the positioning of air masses in the BL or in the FT over a complex terrain such as that of Reunion during the hours preceding their arrival at the site.

Table 1 Overview of the availability of instruments and modelling tools during the OCTAVE (March – May 2018, focus on the period between April 11 and 17) and BIO-MAIDO (March 14 – April 8, 2019) campaigns.

| | Instrument / model | Variable / model prediction | BIO-MAIDO | OCTAVE |
|---|---|---|---|---|
| Instruments | Meteorological station | Wind speed and direction, temperature, RH, radiation | × | × |
| | DMPS | Particle number size distribution | × | × |
| | CI-APi-TOF | Neutral molecule and cluster composition | | × |
| Models | Meso-NH | Boundary layer height | × | |
| | CAT | Air mass backtrajectories | × | × |
| | Meso-CAT | Air mass backtrajectories | × | |

## 3. Identification of a tracer of free tropospheric conditions at Maïdo

### 3.1 Identification of a tracer of free tropospheric conditions

As mentioned in Sect. 2.1.1, the work of Lesouëf et al. (2013) has allowed to characterize the development of the BL on the slopes of Maïdo, highlighting a diurnal cycle characteristic of high altitude sites. The first part of this study aims to complete this work by identifying, with a similar concurrent model-measurement analysis, a tracer allowing to identify FT conditions at

the station from continuous in-situ measurements. The development of the BL in mountainous areas is the result of a complex combination of multiple processes and parameters (De Wekker et al., 2015; Collaud Coen et al., 2018). It should be noted, however, that the objective here is not to study or represent this complexity but to determine a simple tracer allowing a reasonable distinction between FT and BL conditions at the site.

Figure 1 presents the timeseries of the BLH retrieved by Meso-NH for the grid point of the Maïdo observatory (in the innermost domain of the model). The simulated BLH has a marked diurnal cycle, consistent with earlier findings at the site by Lesouëf et al. (2013), as well as with observations from other sites (Rose et al., 2017; Farah et al., 2018; Chauvigné et al., 2019). The model indicates that the station lays in the BL during the day, and predicts in contrast BL thickness values approaching zero during the night. These nightime BL thickness predictions translate into BLH being close to the station elevation in Fig. 1 because the BLH is calculated as the sum of the BL thickness and the grid point elevation (see Sect. 2.2.1). These nightime values more broadly indicate that during the corresponding hours the influence of the BL is likely to be very limited at the site, with, instead, measurements more representative of the FT and associated long-range transport under the influence of a strengthened large –scale subtropical subsidence. In practice, for the rest of the analysis, BL thicknesses below 6 m predicted by the model were associated with FT conditions at the site (i.e. 55% of the time on the 26 days considered during BIO-MAIDO), and the rest of the points with BL conditions. A limitation of this simple classification is that it does not explicitly include the transition regime, during which the site is at the interface between the two layers; due to the strict threshold on the BL thickness used for the classification, however, these transition conditions are likely identified with the periods considered as representative of BL conditions, and therefore should not impact the identification of FT conditions, which remain the focus of this work.

Following the approach of Rose et al. (2017), the standard deviation of the horizontal wind direction ($\sigma_\theta$) was tested first as a potential FT tracer. It should be noted that vertical wind is obviously expected to be a better tracer of vertical turbulence, and therefore of FT and BL conditions, but this variable is not measured at Maïdo. Similarly to Rose et al. (2017) and Chauvigné et al. (2019), higher $\sigma_\theta$ values were assumed to reflect the turbulent conditions associated to an influence of the BL at the site, while lower values were in contrast associated to stable conditions more representative of the FT. As in the abovementioned studies, the calculation of $\sigma_\theta$ was approximated by the Yamartino (1984) single pass method (set of Eq. 1).

$$\sigma_\theta = \sin^{-1}(\varepsilon)\left[1 + \left(\frac{2}{\sqrt{3}} - 1\right)\varepsilon^3\right] \tag{1}$$

where $\varepsilon = \sqrt{1 - (S^2 + C^2)}$

with $S$ and $C$ representing the mean sine and cosine of the wind angle $\theta_i$ (calculated from a sequeunce of N measurements) respectively.

$$S = \frac{1}{N}\sum_{i=1}^{N}\sin\theta_i$$

$$C = \frac{1}{N}\sum_{i=1}^{N}\cos\theta_i$$

To avoid wind-meandering effects (and thus over-estimations of $\sigma_\theta$), the same approach as Rose et al. (2017) was applied: 15 min-averaged values of $\sigma_\theta$ were calculated and further used to derive hourly averages of $\sigma_\theta$ (Eq. 2) that were used to discriminate between FT and BL air masses at the site.

$$\sigma_{\theta(1h)}^2 = \frac{\sigma_{\theta(15)}^2 + \sigma_{\theta(30)}^2 + \sigma_{\theta(45)}^2 + \sigma_{\theta(60)}^2}{4} \tag{2}$$

As evidenced in Fig. 1, the variations of $\sigma_\theta$ strongly correlate with that of the BLH retrieved by Meso-NH, with, as expected, on average higher $\sigma_\theta$ values during daytime under BL influence ($\sigma_\theta = 43 \pm 16$ °) and lower values in the FT ($\sigma_\theta = 16 \pm 13$ °). An exception to this regular diurnal cycle is observed at the end of the investigated period, especially during the night of April 7 to 8, when $\sigma_\theta$ values remain high while the model indicates FT conditions.

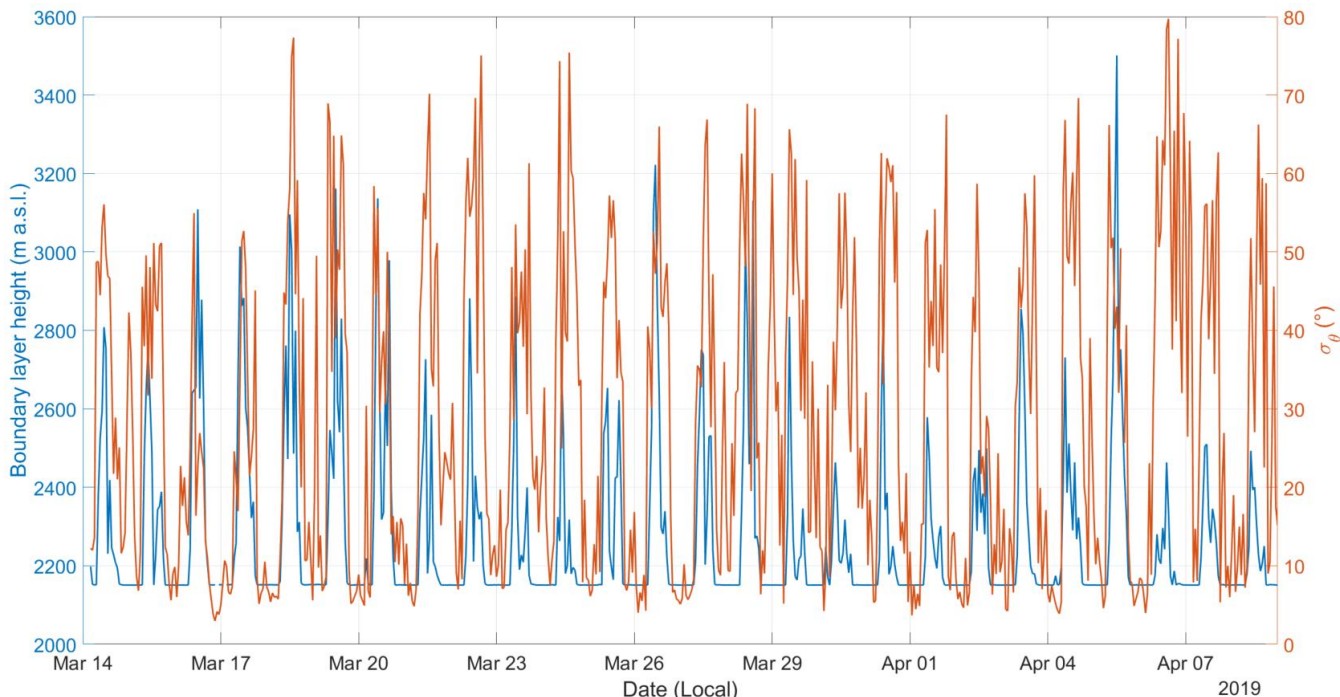

Figure 1 Time series of the BLH (calculated as the sum of the BL thickness and grid point elevation) simulated by Meso-NH for the model grid point which includes the Maïdo observatory in the innermost domain (blue) and the standard deviation of the horizontal wind direction ($\sigma_\theta$) measured at the site (red).

In order to further define a threshold on $\sigma_\theta$ to discriminate the hours when the site is in the BL from the hours when the conditions are more representative of the FT, Fig. 2 presents the percentage of hours when Meso-NH indicates that the station is in the BL / FT as a function of $\sigma_\theta$ values grouped by 2° bin. We observe again that the probability that the site is in the FT is globally higher the lower $\sigma_\theta$ values are. The value of 16°, below which the probability that the site is in the FT is higher than 85% (excluding the 2-4° bin, which contains only 6 points), was defined in this work as the threshold under which the conditions at the station are assumed to be representative of the FT. This threshold is slightly higher than that used for Chacaltaya (12.5°) by Rose et al. (2017) and Chauvigné et al. (2019) who followed the recommendation from Mitchell (1982). However, Fig. 2 indicates that in our case, using the stricter threshold of Mitchell (1982) would have impacted the amount of FT data to be included in the analysis without improving significantly the filtering of BL air masses.

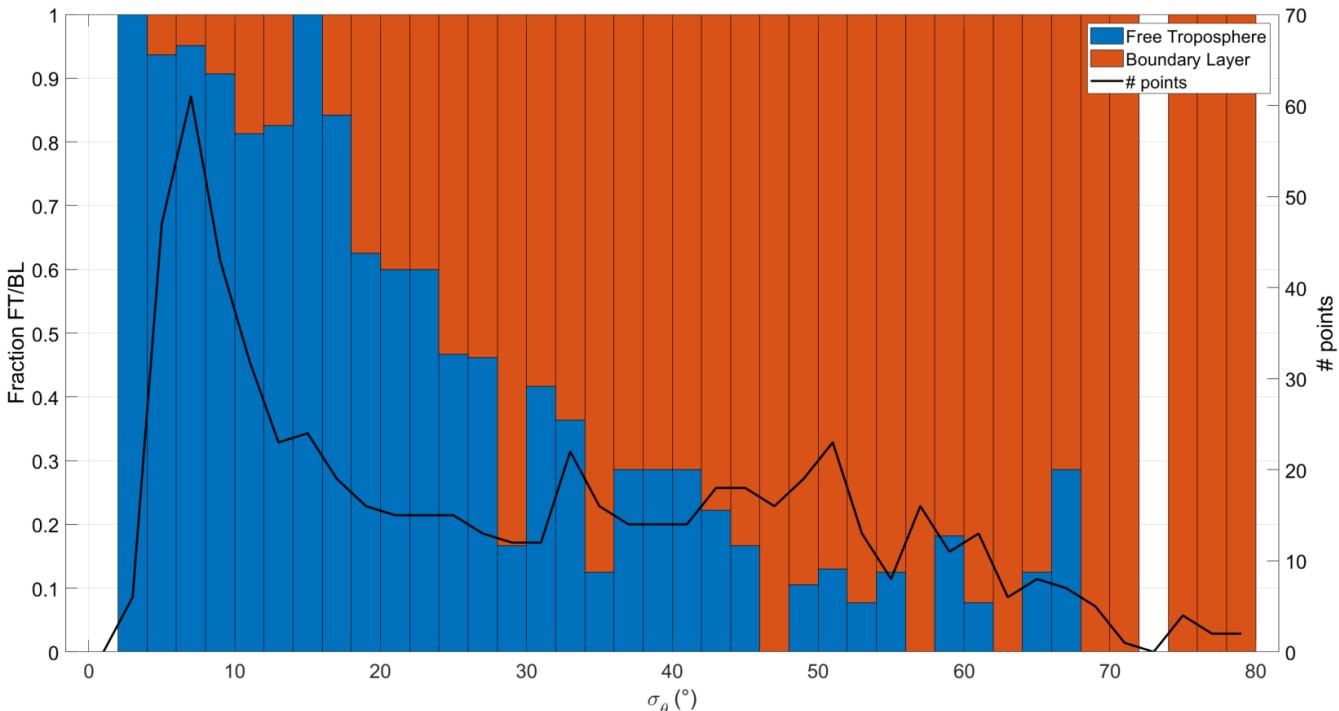

Figure 2 Standard deviation of the horizontal wind direction ($\sigma_\theta$) segregated between FT and BL conditions at Maïdo. The percentage of hours for which the Meso-NH model indicates that the station is in the BL or in the FT for each $\sigma_\theta$ bin (2° width) is shown on the y-left-axis. The number of points in each bin is also represented on the y-right-axis to provide further information on the distribution of the data.

In addition to $\sigma_\theta$, the number concentration of particles > 90 nm ($N_{90}$) was tested as a second potential tracer since it was previously shown to be a good indicator of BL influence at other mountain sites (Herrmann et al., 2015; Farah et al., 2018). Similarly to $\sigma_\theta$, increased occurrence of FT conditions coincides with the lowest $N_{90}$ values (Figs. S2 and S3). In particular, the fraction of FT conditions increases significantly for $N_{90} < 300$ cm$^{-3}$ (> 40%) and is nearly 70% for $N_{90} < 200$ cm$^{-3}$, which is close to FT background concentrations reported for puy de Dôme (~150 cm$^{-3}$; Farah et al., 2018). However, it appears complex to define a threshold on $N_{90}$ allowing to obtain a probability that the station is in the FT as high as the one obtained with $\sigma_\theta < 16°$ without significantly reducing at the same time the number of FT data points. An explanation for this is that $N_{90}$ is not affected only by the dynamics of the BL; indeed, low $N_{90}$ values may be associated to cloud formation (as sub-90 nm particles can likely serve as CCN) while, in contrast, high values can be caused by local pollution. For this reason, $\sigma_\theta$ was used as a tracer of FT conditions in the rest of this study.

It should be noted that in the absence of wind measurements, future studies conducted at Maïdo could simply rely on time ranges to discriminate between FT and BL conditions at the site. In fact, as shown in Figs. 1 and S4, the transition generally takes place at a very regular time, with free tropospheric conditions observed on average between 15:00 and 03:00 UTC (local time - 4 h)). Whenever possible, however, $\sigma_\theta$ should be used, as this parameter reflects the actual atmospheric situation at a given time, and in particular makes it possible to identify periods during which the site is in FT conditions outside the average time window identified.

**3.2 Positioning of the air masses in the FT or in the BL during the hours preceding their arrival at the site**

In addition to the identification of FT conditions at the station, the travel conditions of the air masses (in the FT or in the BL) prior to their arrival at Maïdo in the FT are of great interest to get insight into the sources (of both aerosols and their precursors) that may have influenced the composition of these air masses. As indicated in Table 1, Meso-NH (and therefore Meso-CAT) simulations have not been performed during the OCTAVE period. The possibility of studying the history of air masses arriving in the FT at the site with the less resolved CAT model (which uses ECMWF ERA-5 reanalyses, see Sect. 2.2.2) was therefore

evaluated based on the simulations performed in the framework of BIO-MAIDO, when both Meso-CAT and CAT simulations were available (Table 1).

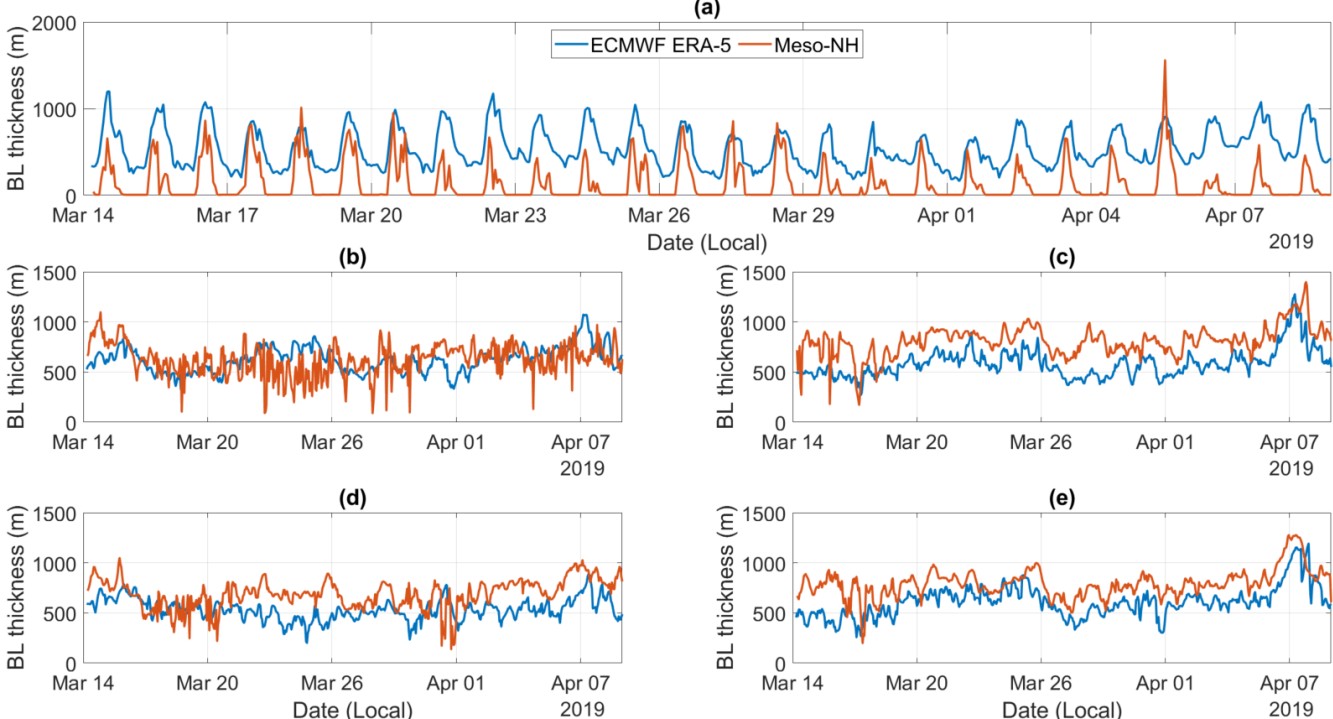

Figure 3 BL thicknesses retrieved by ECMWF-ERA-5 reanalysis and Meso-NH a. at the grid point containing the Maïdo station and above the Indian Ocean to the b. northwest, c. northeast, d. southwest and e. southeast of Reunion island. The exact locations used for the comparisons presented in b-e are shown in Fig. S6.

The BL thicknesses simulated by ECMWF and Meso-NH were compared above the land and above the sea using a selection of points assumed to be representative of these two types of environment. It should be noted that to ensure consistency in the analysis, data from the Meso-NH intermediate domain (500 m horizontal resolution) have been used here since the small domain did not include any sea area. Above land, the grid point containing the Maïdo observatory has been selected in each of the two models to perform the sensitivity test. It can be seen in Fig. 3.a that there are significant differences between the BL thicknesses from the ECMWF ERA-5 reanalyses and those provided by Meso-NH, with systematically higher values associated to ECMWF during the night. In particular, the BL thickness is on average ~383 m and never goes below 163 m in ECMWF ERA-5 reanalyses when Meso-NH predicts FT conditions at the site. These discrepancies are certainly linked to the fact that the topography of Reunion Island is not represented finely within ECMWF (31 km horizontal spatial resolution), which therefore cannot provide robust information on the complex BL dynamics over such a mountainous terrain. Additionally, a deeper analysis of the back-trajectories computed with Meso-CAT was performed, to evaluate the number of hours elapsed over land before the air masses reach the Maïdo in the FT. During the BIO-MAIDO campaign, less than 50% of the air masses arriving from the FT at the site were already over the island 4 hours before reaching the station (Fig. S5.a), and among these air masses, a vast majority (>80%) remained at an altitude higher than that of the station (i.e. certainly in the FT) throughout their journey, including during the last hours over land. (Fig. S5.b). This is consistent with the large-scale subtropical subsidence in which the station is located throughout the night (Baldy et al., 1996 ; Baray et al., 2013). Note that for this analysis, for each of the back-trajectory points along their path, air masses were considered to be over land (or over the ocean) if at least 75% of the 324 trajectories in the set (see Sect. 2.2.2) were over land (or over the ocean), based on Meso-NH topography. Otherwise, their positioning was classified as undefined. Similar approach was applied to assess the positioning of the air masses relative to the station elevation. Over the ocean, 4 points located around the island, in the corners of the Meso-NH intermediate domain, were selected to test adequacy between models (Fig. S6). As shown in Fig. 3.b-e, the BL thicknesses

retrieved by the two models above the sea are overall in better agreement than what was observed above land, with average differences between Meso-NH and ECMWF ERA-5 reanalyses of +192 /-121 m.

In summary, based on the previous results, low resolution models such as ECMWF provide a reasonable prediction of the positioning of air masses relative to the BLH along their path only over the ocean. When predicted to originate from the FT at Maïdo, it is reasonable to make the systematic assumption that air masses were also in the FT during the last hours of their journey over the island.

## 4  Insights into the chemical composition of molecular clusters and their precursors in the marine FT

### 4.1  Overview of the conditions during the OCTAVE campaign

Specific attention was paid to the time period between April 11 and 17, 2018 during which regular conditions were observed at the site (i.e. with no significant influence of tropical storm or volcanic eruption of the Piton de la Fournaise, which is located 39 km south of the station). The overview of the conditions during this time period (previously introduced by Rose et al. (2021)) is presented in Fig. 4.

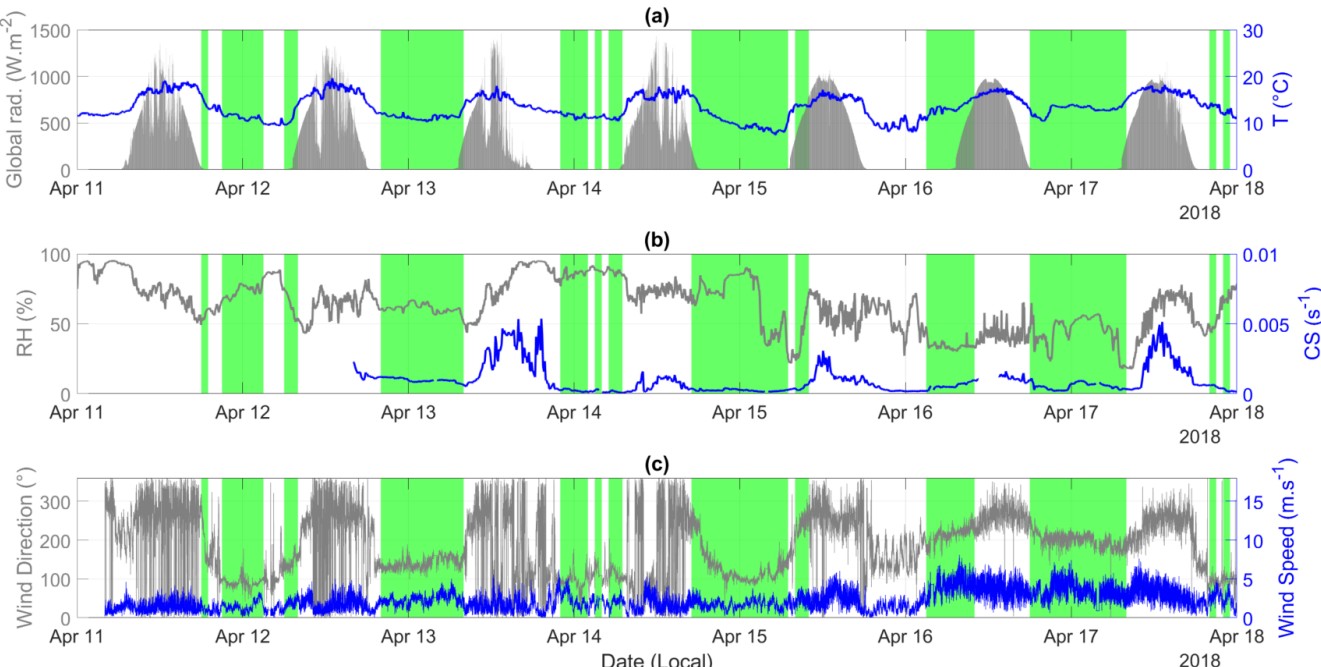

Figure 4 Overview of the meteorological conditions during the OCTAVE campaign conducted in 2018. Green patches depict the hours when the Maïdo station is in the FT based on the analysis of the standard deviation of the horizontal wind direction. Time series of a. global radiation and temperature, b. relative humidity (RH) and condensation sink (CS) and c. wind direction and wind speed.

Based on the analysis of $\sigma_\theta$, the station was found to be 36% of the time in the FT during the 7 days of interest in OCTAVE. In agreement with the results obtained for the BIO-MAIDO campaign in Section 3, FT conditions coincide with nightime, with an increase in global radiation that starts on average about 90 minutes before the BL reaches the station (Fig. 4.a); an exception is observed on the morning of April 16, 2018 when FT conditions remain later at the station until 06:00 UTC (10:00 local time). One should note, however, that although daytime FT conditions could be sampled on that day, it unfortunately did not give the opportunity to study a case of MFT NPF as we could not identify clear evidence of NPF (this day was rather classified as "Undefined" based on the classification of Hirsikko et al. (2007); Fig. S7). Relative humidity is slightly lower in the FT than in the BL (median RH of 61% (10th percentile 34% - 90th percentile 85%) and 67% (43-89%) respectively)(Fig. 4.b), as well as temperature (11.7°C (9.6-13.8°C) in the FT vs 15.4 (10.9-17.4°C) in the BL)(Fig. 4.a), although for temperature there is a bias due to the diurnal variation of the global radiation. As illustrated in Fig. S8, a more pronounced contrast between FT and BL conditions is observed when considering the water mixing ratio instead of RH (7.4 g kg$^{-1}$ (4.3 - 10.5 g kg$^{-1}$) in the

FT vs 8.6 g kg⁻¹ (4.5-11.3 g kg⁻¹) in the BL), although the variations of these two variables logically appear to be strongly correlated. For simplicity, we have chosen to report only the analyses and results involving RH (which is measured directly) in the rest of the study, but it is worth noticing that the associated conclusions were confirmed when considering the water mixing ratio instead. Time series of the wind direction and speed are shown in Fig. 4.c. It is noticeable that the wind direction has a strong diurnal variation with southeast winds during the night, when the Maïdo is in the FT, and westerlies during the day, when the station mostly lays in the BL. From April 16, 2018, however, this cycle is disturbed and the wind speed also starts to become higher compared to the previous days. These changes may already reflect the change in atmospheric circulation caused by tropical storm Fakir that hit Reunion island between April 19 and 24, 2018. The variations of the CS also exhibit a marked cycle (Fig. 4.b), which is linked both to the day night cycle and to the alternation of FT and BL conditions at the site: the CS is logically lower in the FT, i.e. mainly at night (median CS of $3.92 \times 10^{-4}$ s⁻¹ ($1.68 \times 10^{-4}$-$9.61 \times 10^{-4}$ s⁻¹) and $1.07 \times 10^{-3}$ s⁻¹ ($1.79 \times 10^{-4}$-$5.90 \times 10^{-3}$ s⁻¹), in the FT and in the BL, respectively), when particles emitted in the BL are not transported to the site and there is, moreover, no photochemistry promoting the formation of new particles, whose contribution to the CS can be significant during the day. It should be noted that the missing CS values are due to a malfunction of the DMPS or to filtering of the data impacted by local pollution (i.e. related to the activity on site).

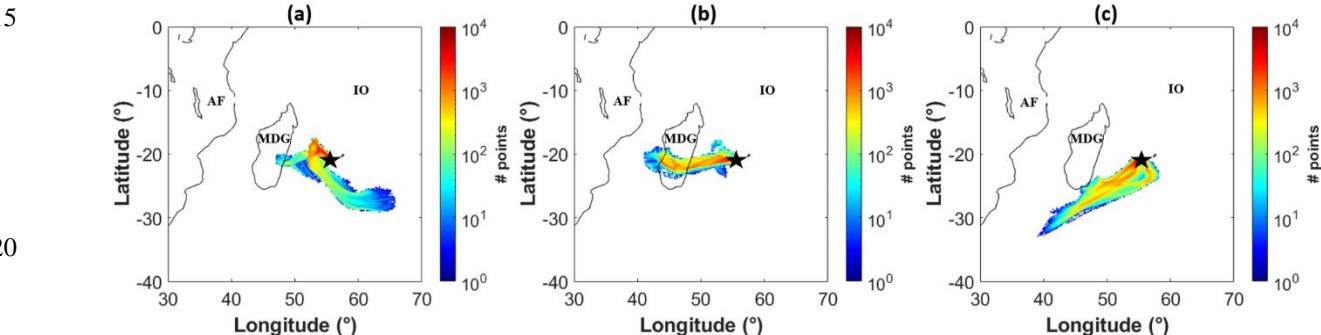

Figure 5 72-hour back-trajectories of the air masses arriving at Maïdo (marked by the black star on the maps) when the station was in FT conditions during the nights of April a. 11 to 12, b. 12 to 13 and c. 14 to 15, 2018. The colour of each grid cell (0.2×0.2°) on the maps indicates the number of back-trajectory points falling into its area. Note that for each back-trajectory computed with the CAT model, all 125 trajectories of the corresponding set are shown in the figure. The abbreviations AF, MDG and IO stand for Africa, Madagascar, and Indian Ocean, respectively.

In order to provide a more detailed description of the air masses arriving in the FT at Maïdo, 72-hour back-trajectories were computed with the CAT model (Sect. 2.2.2 and 3.3). Three main situations were identified during the period of interest, as shown in Fig. 5 which presents the back-trajectories of the air masses reaching the site in FT conditions during the nights of April 11 to 12, 12 to 13 and 14 to 15, 2018 as examples. Most of air masses arriving at the station in the FT are either representative of 1) pristine marine air from the Indian ocean (Fig. 5.a), 2) terrestrial air that has crossed the island of Madagascar, as illustrated in Fig. 5.b for the night of April 12 to 13 and also observed on the nights of 13 to 14, 15 to 16 and 16 to 17 (Fig. S9), or 3) marine air from southern Madagascar (Fig. 5.c).

The altitude of these air masses was also examined to determine whether they were in the FT or in the BL on their path (Fig. S10). The vast majority of the sampled air masses remained in the FT for the last 72 hours before arriving at the station, with only few sporadic passages of some of the trajectories of the computed sets in the BL over the Southern Indian Ocean during the night of 11 to 12, over Madagascar during the night of 13 to 14 and off the southeast coast of Madagascar on the night of 14 to 15. Similar to Sect. 3.2, for each of the back-trajectory points along their path, air masses were considered to be over land (or over the ocean) if at least 75% of the 125 trajectories in the set (see Sect. 2.2.2) were over land (or over the ocean). Unlike in Sect. 3.2, however, in which the positioning of trajectories over land or ocean was studied on the basis of trajectories calculated with Meso-CAT, and where the land-sea distinction was therefore based on the Meso-NH topography, the land-sea mask from the ECMWF ERA-5 reanalyses has been used here. The mask was used in such a way that only the locations associated with a zero mask value were considered to be oceanic; this excludes in particular coastal waters, for which we

assumed that there was a greater probability of terrestrial influence, and therefore chose to classify them with the terrestrial zones. According to this approach, only one of the air masses sampled at Maïdo in the FT during OCTAVE passed through the terrestrial boundary layer (over land other than Reunion; on April 13th), and only one through the marine boundary layer (on April 11th). Therefore a marked terrestrial signature is not expected in the measurements performed in the FT at Maïdo,

which seems to be confirmed by the absence of significant contrasts between the CS values observed on the different nights depending on whether the sampled air masses travelled over land or only sea (Fig. 4.b). Overall, these observations are consistent with the recent results of Mascaut et al (2022), which indicate that nightime measurements performed at Maïdo are mostly representative of the MFT.

### 4.2   Chemical composition of clusters and molecules observed in the marine FT

10       **4.2.1**     **Global overview and contrast with BL observations**

Measurements performed with a nitrate CI-APi-TOF in the framework of OCTAVE were analysed to get insights into the chemical composition of neutral clusters and molecules present in the MFT. Figure 6.a presents the average spectrum (with a unit mass-to-charge resolution, UMR) over the mass-to-charge (*m/z*) range of 80-400 Th measured in FT conditions over the investigated period. Besides nitrate ions and their isotopes, a number of products were identified and classified into 9 groups

according to their chemical composition: fluorinated species (($H(CF_2)_{3-5}COOH$)·$NO_3^-$), iodic acid ($IO_3^-$ and ($HIO_3$)($HNO_3$)$_{0-1}$·$NO_3^-$), sulfuric acid (($H_2SO_4$)$_{0-2}$·$HSO_4^-$ and $H_2SO_4$·$NO_3^-$), $SO_5^-$, MSA ($CH_3O_3S^-$ and ($CH_4O_3S$)($HNO_3$)$_{0-1}$·$NO_3^-$), MSA-derived (including $CH_4O_3S$·$HSO_4^-$ and $CH_4O_3S$·$IO_3^-$), dicarboxylic acids (including oxalic acid, i.e. $C_2HO_4^-$ and $C_2H_2O_4$·$NO_3^-$, malonic acid, i.e. $C_3H_3O_4^-$ and $C_3H_4O_4$·$NO_3^-$, and maleic acid, i.e. $C_4H_3O_4^-$ and $C_4H_4O_4$·$NO_3^-$), C2 (($C_2H_7N$)($HNO_3$)$_{1-2}$·$NO_3^-$) and C4 (($C_4H_{11}N$)($HNO_3$)$_{1-2}$·$NO_3^-$) amines. The identified species belonging to these groups are highlighted in Fig. 6, and further listed

with their exact mass and assumed composition in Table 2. In order to offer a different perspective on the data, Fig. S11 presents in addition the mass defect plot (corresponding to the data presented in Fig. 6.a) highlighting the abovementioned species as well as the signals of the non-identified compounds associated to major peaks which are briefly discussed at the end of the section.

As shown in Fig. 7.a, the identified species represent on average 21.2% of the total signal over the *m/z* range of 80-400 Th,

with, on average, very similar fractions in the FT and in the BL (22.1% and 20.5%, respectively); note that the reagent ions (including as well the water cluster $H_2O$·$NO_3^-$) and their isotopes, which signals are obviously much higher than those of the other compounds present in the spectrum, have been excluded from Fig. 7 (they are not considered either in the calculation of the total signal or in the calculations related to the identified fraction). Also, to make them easier to read, the results shown in Fig. 7 are averaged over 30 minutes; the fractions indicated in the text, on the other hand, were calculated on the 3-minute averaged signals to retain the information associated with this higher temporal resolution. In addition to Fig. 6.a, Fig. 6.b

shows, over the same *m/z* range, the difference between the average spectra measured in FT and BL conditions. Since the segregation between BL and FT conditions is very strongly coupled to the day night cycle, the interpretation of the results in Figure 6.b, and in particular the evidence of a signal variability related to the BL dynamics, requires a particular precaution which we have taken in the rest of this section. This also limits the possibility of studying the involvement of the compounds

identified in the FT in daytime NPF. For this reason, as mentioned before, we have not addressed this aspect here, which may be the subject of a future study based on a longer dataset allowing a higher probability of collecting daytime measurements in the FT. When it has been highlighted in the literature, however, the importance of the identified compounds with regard to particle nucleation and/or growth processes is mentioned.

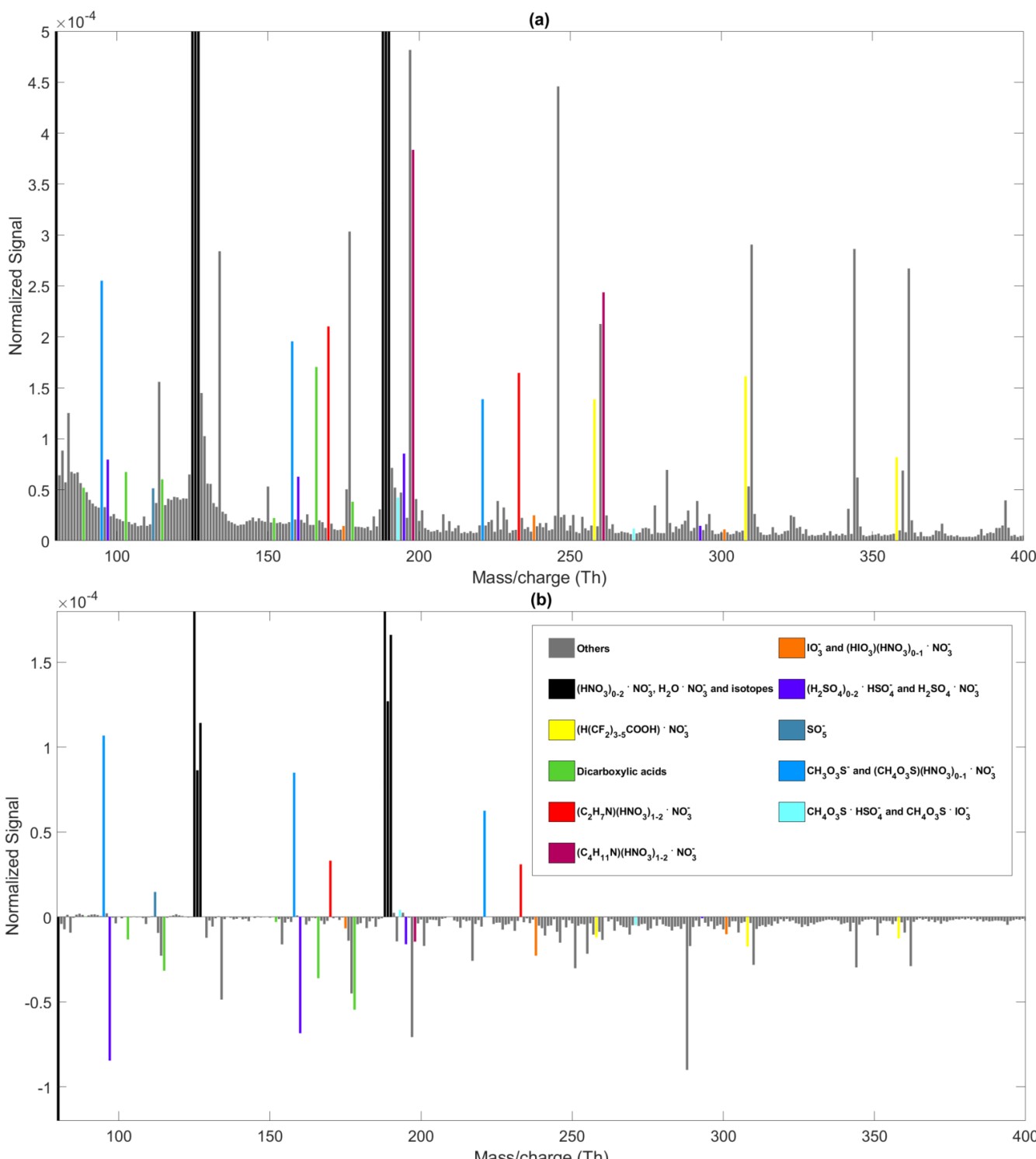

Figure 6 a. Average spectrum (in the *m/z* range of 80-400 Th) obtained with nitrate CI-APi-TOF in FT conditions at Maïdo (measurements performed between April 11 and April 17) and b. difference between the average mass spectra measured in the the FT and in the BL. The signals are normalised by the sum of the reagent ion (i.e. $NO_3^-$, $HNO_3 \cdot NO_3^-$, $(HNO_3)_2 \cdot NO_3^-$) signals. The species belonging to considered series are highlighted with a colour code indicating the group to which they belong, and are further listed, with their exact mass, in Table 2.

Table 2 List of molecules and clusters identified in the FT at Maïdo. The last column shows the assumptions resulting from the analysis reported in Sect. 4.2.2 concerning the origin of the compounds.

| Group | Exact molecule or cluster mass (Th) | Assigned formulae | Hypothesized source |
|---|---|---|---|
| Fluorinated species | 257.984 | $H(CF_2)_3COOH \cdot NO_3^-$ | Contaminants emitted by fluorinated plastic (tubing used in the CI-inlet setup) |
| | 307.981 | $H(CF_2)_4COOH \cdot NO_3^-$ | |
| | 357.978 | $H(CF_2)_5COOH \cdot NO_3^-$ | |
| Iodic acid | 174.890 | $IO_3^-$ | Marine |
| | 237.885 | $HIO_3 \cdot NO_3^-$ | |
| | 300.881 | $(HIO_3)(HNO_3) \cdot NO_3^-$ | |
| Sulfuric acid | 96.960 | $HSO_4^-$ | Marine (oxidation product of DMS) |
| | 159.956 | $H_2SO_4 \cdot NO_3^-$ | |
| | 194.927 | $(H_2SO_4) \cdot HSO_4^-$ | |
| | 292.895 | $(H_2SO_4)_2 \cdot HSO_4^-$ | |
| $SO_5^-$ | 111.947 | $SO_5^-$ or $O_2 \cdot SO_3^-$ | Marine |
| MSA | 94.981 | $CH_3O_3S^-$ | Marine (oxidation product of DMS) |
| | 157.976 | $(CH_4O_3S) \cdot NO_3^-$ | |
| | 220.972 | $(CH_4O_3S)(HNO_3) \cdot NO_3^-$ | |
| MSA-derived | 192.948 | $CH_4O_3S \cdot HSO_4^-$ | Marine |
| | 270.878 | $CH_4O_3S \cdot IO_3^-$ | |
| Dicarboxylic acids | 88.988 | $C_2HO_4^-$ | Marine and remote terrestrial |
| | 151.984 | $C_2H_2O_4 \cdot NO_3^-$ | |
| | 103.004 | $C_3H_3O_4^-$ | Remote terrestrial |
| | 165.999 | $C_3H_4O_4 \cdot NO_3^-$ | |
| | 115.004 | $C_4H_3O_4^-$ | Marine |
| | 177.999 | $C_4H_4O_4 \cdot NO_3^-$ | |
| C2 amines | 170.042 | $(C_2H_7N)(HNO_3) \cdot NO_3^-$ | Marine and nearby terrestrial |
| | 233.038 | $(C_2H_7N)(HNO_3)_2 \cdot NO_3^-$ | |
| C4 amines | 198.073 | $(C_4H_{11}N)(HNO_3) \cdot NO_3^-$ | Suspected contamination |
| | 261.069 | $(C_4H_{11}N)(HNO_3)_2 \cdot NO_3^-$ | |

As shown in Fig. 7.b, the fluorinated compounds highlighted in Fig. 6 constitute a non-negligible fraction of the identified signal of 18.2%, with no marked difference between BL and FT conditions (on average 18.3% and 17.9%, respectively). Since blanks were not performed during this campaign due to the complex experimental setup position, the origin of these compounds remains uncertain but, similar to Ehn et al. (2012), we suspect that these molecules are contaminants emitted by fluorinated plastic (i.e. polytetrafluoroethylene, PTFE, in this case), present in our case in the tubing used in the CI-inlet setup. Only three unambiguously identified fluorinated compounds are highlighted here for this group although it is likely that other compounds of this family are present in the spectra, at a span corresponding to multiples of the mass of the $CF_2$ group (unit mass of 50 Th); however, we have not carried out an in-depth analysis for the identification of these compounds as they most probably do not have an atmospheric origin. The three identified peaks were nevertheless used for the mass calibration of the data since, as mentioned previously, these species have the benefit of being present during the day and night with a small diurnal cycle and little overall variability over the period of interest (Fig. S12).

Sulfuric acid, which is commonly accepted as a key precursor of NPF in a variety of environements including clean marine air (Beck et al., 2021; Baccarini et al., 2021; Brean et al., 2021; Peltola et al., 2022), is observed in the spectra (Fig. 6.a). As shown in Fig. 6.b, the signals associated with the products in this group are higher in the BL, and the resulting concentrations are on average of the order of $4.69 \times 10^6$ cm$^{-3}$ compared to $1.75 \times 10^6$ cm$^{-3}$ in the FT (Fig. 8.a). The sulfuric acid group also represents a higher fraction of the total signal identified in the BL than in the FT (Fig. 7.b, on average 10.3% and 3.2%, respectively), up to 63.2% on April 12. This is consistent with a predominant photochemical production of this species, which, in absence of volcanic eruption of the piton de la Fournaise, likely mainly results from the photo-oxidation of $SO_2$ originating from the oxidation of DMS emitted by oceanic phytoplankton, and possibly from anthropogenic activity on the island. Increased sulfuric acid concentrations (up to $2.49 \times 10^7$ cm$^{-3}$) are however observed on the night of April 14-15 (Fig. 8.a) and coincide with a drop in RH (Fig. 4.b). Similar observations were reported by Frege et al. (2017) at the high altitude station of Jungfraujoch (3454 m a.s.l., Switzerland) and Mauldin III et al. (1999) from airborne measurements over the Pacific Ocean. Based on the work of Tsagkogeorgas et al. (2017), the connection between increased sulfuric acid concentration and lower RH might be explained by the evaporation of sulfuric acid from the condensed phase, which they report as the main driver of sulfur particle shrinkage at low RH combining chamber experiments and model simulations.

Besides sulfuric acid, another molecule of importance identified in the spectrum is iodic acid. Evidence for the contribution of iodine-containing compounds to particle formation in the MBL was first reported in coastal areas (e.g. O'Dowd et al., 2002), and later over non-tidal Mediterranean waters (Sellegri et al., 2016). The nucleating potential of iodic acid in particular has since been identified in coastal areas (Sipilä et al., 2016), in open South Pacific Ocean air masses (Peltola et al., 2023), and over the ice pack (Baccarini et al., 2020; Beck et al., 2021). Atmospheric iodine mainly originates from oceans, as a result of both biological processes and sea surface chemistry (Carpenter et al., 2021); iodine can be emitted in different forms, which release the iodine atoms upon photolysis with distinctive efficiencies, from which iodic acid is thought to be formed in the presence of ozone (He et al., 2021; Finkenzeller et al., 2022). Autocatalytic release of iodine from particles has also recently been documented at Mace Head mesurement station (Tham et al., 2021), yet it is currently unclear if this mechanism could explain the observation of iodine species in the MFT. As in the case of sulfuric acid, there is a diurnal variation in the amplitude of the signals of the observed products in the iodic acid group, with on average higher values during daytime (Fig. 6.b) which logically translate into higher iodic acid concentrations in the BL than in the FT (Fig. 8.a, on average $7.19 \times 10^5$ cm$^{-3}$ and $2.90 \times 10^5$ cm$^{-3}$, respectively). This also leads to a contribution to the total signal identified which is higher in the BL than in the FT (Fig. 7.b, on average 2.5% and 1.0%, respectively), although significantly lower than that of the sulfuric acid group. These observations are again consistent with a predominant photochemical production, but the variations observed for iodic acid are different from those observed for sulfuric acid, with increased concentrations during sunrise and sunset and a significant decrease at noon, especially on April 12 and 14 (Fig. 8.a). This peculiar behaviour was previously reported in other studies performed in the MBL (Sellegri et al., 2016; Baccarini et al., 2021; Peltola et al., 2023) and was also observed at Jungfraujoch (Frege et al., 2017), and is suspected by Baccarini et al. (2021) to be related to lower iodic acid formation yield at higher solar irradiance. However, since similar variations were observed for sulfur compounds on April 14 (Fig. 8.a and d), it may be that specific conditions at the site simultaneously influenced the variations of all these species on that day; deeper investigation of this event observed in BL conditions is nevertheless beyond the scope of this work. In addition, as illustrated in Fig. 8.a, the morning increase of the iodic acid concentration starts ~ 1 hour before that of sulfuric acid, when the station is still in the FT. This is consistent with the results of He et al. (2021) derived from chamber experiments which suggest that iodic acid formation can be triggered under low radiation levels, and whose relevance for real atmospheric conditions has since been demonstrated by Finkenzeller et al. (2022) using measurements performed at Maïdo during OCTAVE.

The MSA group shows a distinct pattern from the previous two groups, with signals that are on average significantly higher in the FT than in the BL (Figs. 6.b and 8.a) and can represent up to 61.2% of the total signal identified in the FT (17.7% on average, compared to 9.4% in the BL). MSA is, with sulfuric acid, the second most considered oxidation product of DMS. The

ability of MSA to form particles (with amines and ammonia) has been shown in flow reactors (Chen et al., 2016, 2017). Using quantum chemical calculation and the Atmospheric Cluster Dynamics Code (ACDC; McGrath et al., 2012) to simulate cluster growth, its potential to accelerate the nucleation of sulfuric acid and dimethylamine was reported by Bork et al. (2014) and its ability to promote iodic acid cluster formation was more recently highlighted by Ning et al. (2022). Yet, field observations suggest that MSA may contribute more to particle growth than to their formation (Berresheim et al., 2002; Baccarini et al., 2021; Brean et al., 2021; Beck et al., 2021; Peltola et al., 2023). Although the MSA group displays overall higher signals in the FT, MSA concentration has no marked diurnal cycle at Maïdo (Fig. 8.a), in line with earlier observations in the MBL (Berresheim et al. 2002; Baccarini et al. 2021; Peltola et al., 2023) and from the continental mountain sites of Jungfraujoch (Frege et al., 2017) and Chacaltaya (Scholz et al., 2023), and significantly higher concentrations are in particular observed during the second half of the investigated period. Similar to Berresheim et al. (2002) and Baccarini et al. (2021), the signal of this group seems to vary rather much with RH, with increased values under dryer conditions which may reflect the evaporation of MSA from the condensed phase (Baccarini et al., 2021). As also pointed out by Scholz et al. (2023) at Chacaltaya, MSA concentrations are in addition favoured by lower CS values at Maïdo, which are overall characteristic of the FT (Fig. 4.b). Using a linear regression model, RH and CS are found to explain 51.6% of MSA variability at Maïdo over the whole dataset, and 83.8% when selecting only the periods during which the site is in FT conditions (Fig. S13). Like at Jungfraujoch (Frege et al., 2017), MSA was also observed in clusters with sulfuric acid and iodic acid at Maïdo (Fig. 6), but those were distinguished from the MSA group due to their distinct time traces. The signals associated with these clusters do not show a strong contrast between the FT and the BL (Fig. 6.b), probably due to the fact that they are composed of species that behave in opposite ways with respect to the BL/FT contrast, and, as shown in Fig. 7.b, they represent on average only a very small part of the identified signal (0.5%).

Besides sulfuric acid and MSA, $SO_5^-$ is another sulfur-containing species identified in the spectrum. The signal contribution associated with this group, which actually likely includes $SO_5^-$ and/or $O_2 \cdot SO_3^-$ (Frege et al., 2017 and references therein), is low overall (on average 1.4% and 0.9%, in the FT and in the BL, respectively) and, although it tends to have a higher signal in the FT (Fig. 6.b), as for MSA there is no marked diurnal cycle for this group. As at Jungfraujoch (Frege et al., 2017), there is a very strong correlation between the signals of the MSA and $SO_5^-$ groups ($R^2$ = 0.98 and 0.97, in the FT and in the BL, respectively; Fig. S14.a). In contrast with Jungfraujoch (Frege et al., 2017) and Chacaltaya (Zha et al., 2022), there is however no single marked correlation with the sulfuric acid group signal during the day (i.e. when global radiation > 10 Wm$^{-2}$), but apparently two correlation regimes depending on the magnitude of the sulfuric acid group signal (Fig. S14.b). Note that for consistency with the abovementioned studies, distinction between day and night was used here because although the BL/FT distinction is closely related to the day night cycle, there are periods when the station is a priori influenced by the BL at night (Fig. 4.a), when sulfuric acid concentrations are otherwise low compared to daytime levels (Fig. 8.a). These observations support the hypothesis of multiple formation pathways for the products in the $SO_5^-$ group. The oxidation of $SO_2$ by $CO_3^-$ or $O_3^-$ (Möhler et al., 1992; Salcedo et al., 2004), which is supported by the observations form the chamber experiments performed by Schobesberger et al. (2015), is a suggested formation route, together with the deprotonation of $HSO_5$, as proposed by Frege et al. (2017) although previously excluded by Ehn et al. (2010). Finally, given the very high correlation observed between the signals of the MSA and $SO_5^-$ groups, we cannot rule out the possibility that $SO_5^-$ is a fragment of MSA in the mass spectra.

Along with MSA and $SO_5^-$, C2 amines are also detected with a clearly higher average signal in the FT than in the BL at Maïdo (Fig. 6.b), and they contribute significantly to the identified signal fraction (Fig. 7.b, on average 16.8% and 13.3%, in the FT and in the BL, respectively). There are about 150 gaseous amines identified in the atmosphere, but on a global scale little is known about the flux of most of them. They can originate from various anthropogenic (e.g. animal husbandry or sewage) or natural sources (Ge et al., 2011a). Oceans are among the known natural sources, with amines possibly resulting from excretion and metabolism from a variety of marine organisms (Wang and Lee, 1994; Calderón et al., 2007). 30% of the three most abundant methylamines (i.e. monomethylamine, dimethylamine and trimethylamine), which estimated cumulative fluxes

nevertheless remain two to three orders of magnitude lower than that of ammonia, could in particular originate from oceans (Schade and Crutzen, 1995). Once in the atmosphere, amines may be subject to various processes and/or interactions, including partionning into the particle phase via direct dissolution (amines are highly soluble) or acid-base chemistry (some are stronger bases than ammonia), oxidation reactions with OH and, although less efficient, with NOx and $O_3$, as well as surface deposition (Ge et al., 2011a, b). As a result, concentrations of amines in the gas-phase are expected to be small (typically not exceeding a few tens of pptv), and in general much lower than that of ammonia, making measurements of gaseous amines in the background atmosphere more challenging, and in turn limited compared to ammonia (Ge et al., 2011a). However, despite their low concentrations, amines are suspected to contribute to nucleation more efficiently than ammonia (e.g. Almeida et al., 2013) and their measurement in the gas phase has therefore recently received considerable attention. The possibility to use a nitrate CI-APi-TOF for sensitive measurement of dimethylamine was demonstrated by Simon et al. (2016) and, the same year, Kürten et al. (2016) reported measurements of C1-C6 amines at the rural site of Melpitz (Germany) using a similar instrument; recently, Brean et al. (2021) have reported nitrate CI-APi-TOF measurements of C2 and C4 amines in the Antarctic Peninsula. The C2 amine signals measured at Maïdo are higher than those reported by Brean et al. (2021) in the MBL (i.e. closer to the ocean, which is one of the identified sources of amines), but they show a similar diurnal cycle, with a minimum during daytime (Fig. 8.b). In contrast, while similar behaviour is observed by Brean et al. (2021) for C2 and C4 amines, there is no marked diurnal cycle in C4 amines signal at Maïdo (which constitute an even larger fraction of the identified signal, on average 28.6% and 27.2% in the FT and in the BL, respectively, Fig. 7.b); sporadic peaks, which are further discussed below, are observed instead. The observations of Kürten et al. (2016) are different, with an absence of diurnal cycle for C2 amines, and a marked daily peak for C4 amines that correlates with temperature evolution and may reflect evaporation from the condensed phase. Brean et al. (2021) attribute the daytime minimum to clustering with sulfuric acid, but such clusters are not observed at Maïdo (nor in Melpitz). This does not preclude the possibility that amines are involved in the daytime formation of sulfuric acid clusters at Maïdo (and Melpitz) but we may hypothesize that the warmer temperatures encountered at these sites compared to the Antarctic Peninsula favour the evaporation of the amines after charging of the sulfuric acid clusters. Another specificity of the Maïdo observations is that, in contrast to the other two sites, the C4 amines group signals are overall higher than those of the C2 amines group (Figs. 8.b). As shown in Fig. S11, peaks consistent with the presence of C6 amines were also observed during OCTAVE; however, given the weakness of the signal these are not further discussed here.

Using a global model, Yu and Luo (2014) investigated the distribution of the 3 most common methylamines on a global scale; their results suggest that in the area of interest for our study, the lifetime of dimethylamine in the gas phase should be in the order of 5 to 10h, depending on the magnitude of the aerosol uptake, with concentrations <0.01 ppt, i.e. considered as a contamination level in the CLOUD experiments (Almeida et al., 2013). Given the expected limited lifetime and concentrations of gaseous amines, amine-driven nucleation is not expected to be efficient at high altitudes. On the other hand, the contribution of the process to CCN concentrations should also be minimal in the vicinity of highest amine sources, which are found in polluted areas and are therefore collocated with elevated CS (which tend to suppress nucleation). For these reasons, amine-driven nucleation is not included in the global model of aerosol formation developed by Dunne et al. (2016). It is however important to note that in the absence of detailed emission inventories, fixed amines to ammonia ratios were used in these two modelling studies to estimate amine emissions, which may lead to uncertainties in predicted concentrations; the concentrations simulated by Yu and Luo (2014) were in particular found to be often significantly lower than the measured values to which they are compared in this same study.

In absence of dedicated instrument calibration and blanks, the actual concentration associated with the amines signal measured at Maïdo could not be assessed, which unfortunately limits our analysis. In the absence of blanks, in particular, the risk of a contamination in the measurements cannot be excluded. In the case of C4 amines, it appears that the peaks mentioned above are in fact similar to those present in the signal of the identified fluorinated compounds (Fig. S12), and more generally there is a strong correlation between the signal of C4 amines and that of $H(CF_2)_4COOH \cdot NO_3^-$ ($R^2 = 0.60$) and $H(CF_2)_5COOH \cdot NO_3^-$

($R^2 = 0.70$). The suspicion of a contamination is therefore high for these compounds, which will not be investigated further. For C2 amines, in contrast, there is no such correlation with fluorinated compounds, and there is also no correlation with temperature (Fig. 10), which further rules out the possibility that these compounds are the result of a temperature modulated contamination by the instrument or the sampling line (volatilization-related). We also ruled out the possibility of an interference between the signal of the cluster composed of $NO_3^-$ and 6 water molecules (mass 170.052 Th) and that attributed to the C2 amine cluster $(C_2H_7N)(HNO_3)_1 \cdot NO_3^-$ (170.042 Th) reported by Kürten et al. (2016) at high RH. Indeed, the time series of the signals associated with the two peaks in the C2 amines group show the same variability (figure not shown), and the signal attributed to the cluster with nitrate dimer is furthermore anti-correlated with RH ($R = -0.18$), and even more with the water mixing ratio ($R = -0.42$). On the other hand, we cannot exclude the presence of a constant background signal (for e.g. caused by the use of a compressor for the generation of the sheath air used in the CI unit), on which would be superimposed a "real" ambient signal determining the observed variations. In this case, however, the amplitude of the reported signals would obviously be overestimated (and as would be that of the associated fractions shown in Fig. 7), but the conclusions related to signal variability would be preserved. We therefore believe that despite the above-mentioned limitations and associated uncertainties, our observations are interesting and should call for further measurements of gaseous amines in the MFT as they may question the existence of an unidentified source.

In addition to the abovementioned compounds, oxalic, malonic and maleic acids are seen in the spectrum (Fig. 6). Since all three are dicarboxylic acids and their signals share common characteristics, they have been classified in the same group although they have specificities. Oxalic acid has signals that are up to an order of magnitude lower than those of the other compounds in the group, and has an overall less pronounced diurnal cycle than malonic and maleic acids, which have clearly higher daytime concentrations (Figs. 6.b and 8). For malonic acid, there is also a persistence of high FT concentrations on the night of 13-14, and an increase in FT concentrations on the night of 16-17, which explain the large signal fraction (up to 44.5% on the night of 16-17) associated with this group during these two time windows. Malonic acid was evidenced at Jungfraujoch (Frege et al., 2017), and more recently at Chacaltaya (Zha et al., 2023) and in clean marine air masses (Peltola et al., 2023), while oxalic acid was also reported at Jungfraujoch (Frege et al., 2017). In marine areas, dicarboxylic acids can be transported from polluted continental regions (see Kawamura et al., 2016 and references therein) but the work of Bikkina et al. (2014) suggests that dicarboxylic acid levels over the open ocean could be controlled by biological activity, and more specifically by in-situ production from marine biogenic emissions of isoprene and subsequent photochemical oxidation.

As illustrated in Figs. 6 and 7 and mentioned above, a significant part of the signal measured with the CI-APi-TOF has not been identified, of which a certain fraction may be made up of organic compounds, and given their mass range in particular highly oxygenated molecules (HOMs). An attempt to identify the compounds observed in Hyytiälä, the birthplace of HOM studies located in the boreal forest (Ehn et al., 2012; 2014), was however unsuccessful. In particular, the molecules that were used as fingerprint for the study of HOM sources by Yan et al. (2016), are not detected at Maïdo (see Table S1 in the Supplement for a list of these compounds). A first explanation for the absence of these HOMs at Maïdo is certainly linked to the fact that their precursors, monoterpenes, are present at low concentrations, below instrumental detection limits during the night (Rocco et al., 2020). Additionally, the meteorological conditions (mainly temperature and relative humidity) and oxidant concentrations, which all certainly affect the chemical reactivity, are very different at Maïdo compared to Hyytiälä. Figure 6 also highlights a certain number of peaks associated with high signals which have not been identified, in particular at UMR 114, 134, 177, 197, 246, 260, 310, 344 and 362 Th. Since these compounds do not seem to be enriched in the FT (Fig. 6.b), they have not been the subject of in-depth study, but the time series of their signals are nonetheless shown in Fig. S15. The signals associated with UMR 114, 134, 177 and 197 Th (Fig. S15. a) show similar variations. Given the difference of 63 Th (corresponding to $HNO_3$), masses 114 and 177 Th on the one hand, and 134 and 197 Th on the other, are most likely associated to the same compound, therefore explaining the similarities within each pair, and the similitude between the two pairs is certainly more broadly indicative of a common source. The other masses (UMR 246, 260, 310, 344 and 362 Th) have a different

behaviour (Fig. S15.b), which is actually comparable to that observed for the peaks attributed to C4 amines (Figs. 8 and 9) and fluorinated compounds (Fig. S12), suggesting that these compounds could once again be the result of a contamination. In particular, as illustrated on the mass defect plot shown in Fig. S11, the peaks identified at UMR 260 and 310 Th have a mass difference corresponding to the mass of the $CF_2$ group, suggesting that these compounds may actually be fluorinated species. These observations may be the subject of more in-depth investigations in future studies dedicated to the BL composition.

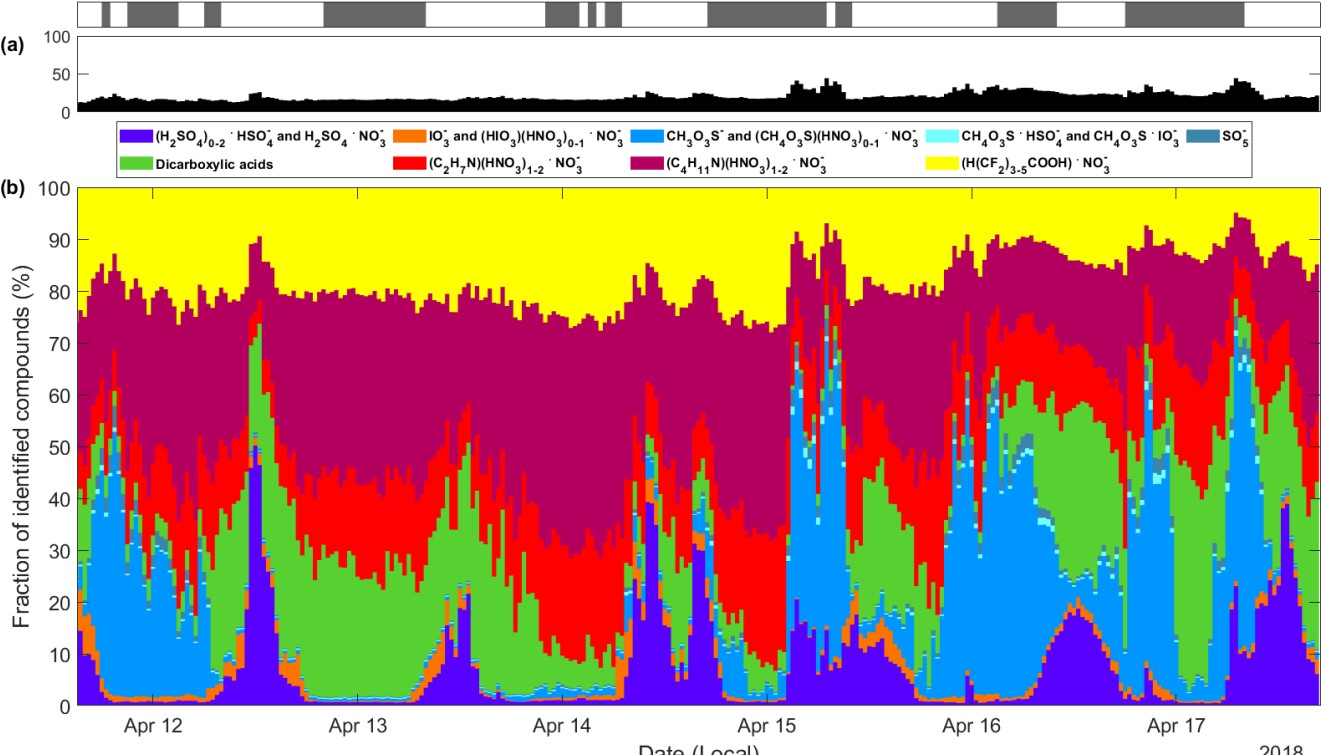

Figure 7 Timeseries of a. the identified fraction of the signal in the *m/z* range of 80-400 Th and b. the contributions of the individual groups to the total identified signal (30 minutes average). The reagent ions (including as well the water cluster $H_2O \cdot NO_3^-$) and their isotopes, whose signals are obviously much higher than those of the other compounds present in the spectrum, are not considered either in the calculation of the total signal or in the calculations related to the identified fraction. The top panel indicates in addition the periods when the station is in FT conditions (gray patches).

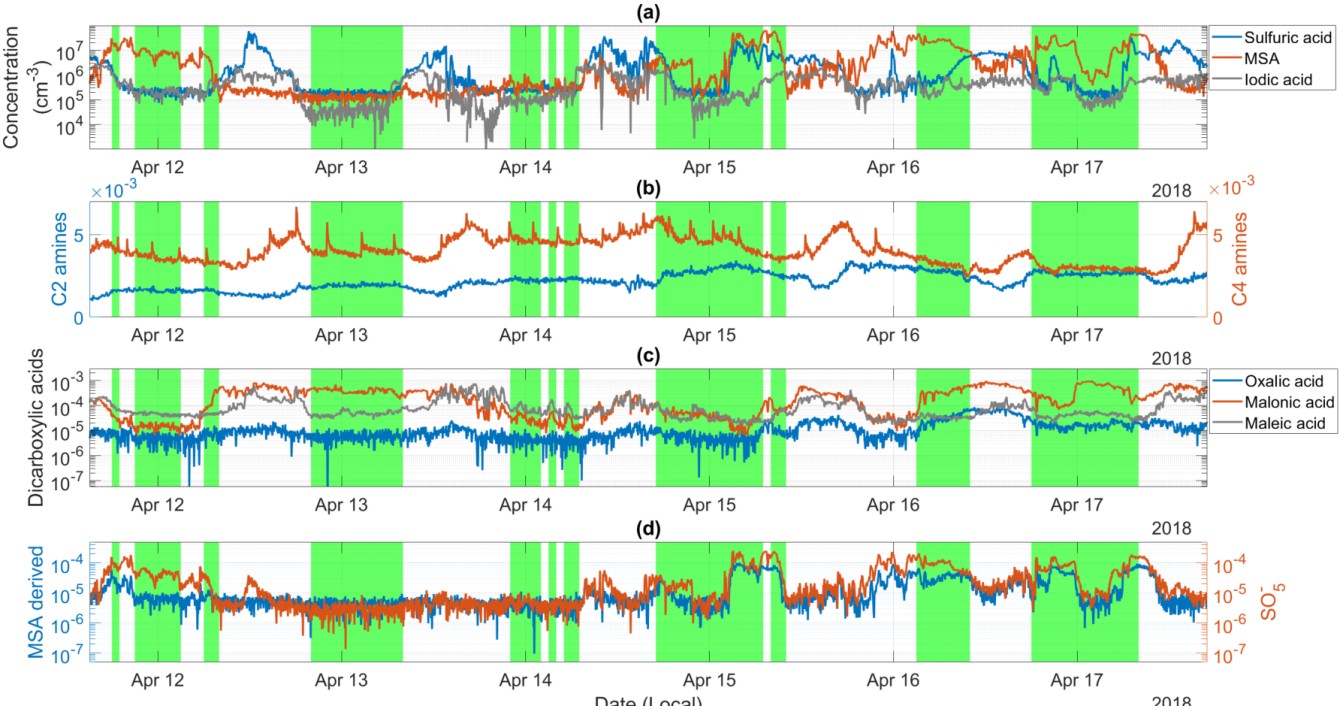

Figure 8 Timeseries of a. sulfuric acid, MSA and iodic acid concentrations, b. normalized amines signal, c. normalized dicarboxylic acids signal and d. normalized MSA-derived and $SO_5^-$ signals. As a reminder, the amines signal are normalized by the nitrate trimer $((HNO_3)_2 \cdot NO_3^-)$ signal in this figure, following the recommendation of Simon et al. (2016). Green patches depict the hours when the Maïdo station is in the FT based on the analysis of the standard deviation of the horizontal wind direction. The timeseries of sulfuric acid concentration has previously been reported in Rose et al. (2021) and that of iodic acid, although over a shorter period, in Finkenzeller et al. (2022).

### 4.2.2 Analysis of the signal variability of the identified compounds in FT conditions

The purpose of this last section is to further investigate the variability of the identified signals specifically in the FT. As illustrated in Fig. 9, when selecting only FT periods we observe that the C2 amine signals increase over the measurement period. Sulfur compounds, iodic acid and oxalic acid show stable and lower overall abundances at the start of the period, with a greater variability in the second half, particularly between the 15th and 16th and between the 16th and 17th. The maleic acid signal is uniform over the whole period, whereas malonic acid shows more variation, with weaker signals overall on the nights of 11 to 12, 13 to 14 and 14 to 15. The very similar trends observed for all the sulfur compounds and iodic acid are confirmed by the strong correlations between their signals (Fig. S16; R = 0.71 – 0.99). They form the purely marine species group that we will discuss as a whole in the following. Here we note that in our data set, sulfuric acid is well correlated to MSA (R = 0.74) and does not seem to contain a significant contribution from anthropogenic sources such as ship emissions. Confirming similarities observed in Fig. 9, we also find a moderate correlation between the purely marine species with oxalic acid (R = 0.24 – 0.45) and C2 amines (R = 0.34 – 0.55), indicating that oxalic acid and C2 amines likely have at least partially a marine origin. Oxalic acid shows in addition a pronounced correlation with malonic acid (R = 0.67). Maleic acid does not seem to vary like any of the other species detected but have a rather individual behavior (Figs. 9 and S16).

Next, we further investigate the factors possibly contributing to the identified compound's variability in the FT. A series of variables were considered, including both measurements concurrently performed at the station (temperature and RH) and variables related to air mass history and provided by ECMWF ERA-5 reanalyses. In addition to the mean radiation (calculated from values > 10Wm⁻²), the number of hours with a non-zero liquid water content (LWC) along the air mass path was considered to highlight a possible effect of wet scavenging in cloud, and the mean RH calculated on the remaining points (i.e. those with zero LWC) was also included in the analysis to study the effect of humidity outside cloudy areas. Note that due to

the coarse horizontal spatial resolution in ECMWF ERA-5 reanalyses (31 km), most convective systems are not represented in the model. In order to have more indications on the origin of the compounds (marine or terrestrial), the influence of the time spent by the air mass over the ocean since it last passed over land was also studied. It should be noted that the air masses having never flown over a land other than Reunion (during the 72h of back-trajectory considered) were not included in the correlation analysis with this last parameter (because the exact time since their last passage over land is unknown). The time spent over the island was analysed instead for these particular air masses to provide a possible additional indication on the variability of the associated signals, and identify in particular a possible nearby terrestrial origin for the observed compounds. The underlying assumption is that the longer the time spent by the air mass over the island, the more likely it is that a nearby terrestrial source may have impacted the observations at Maïdo through exchanges between the BL and the FT. Finally, the total time spent over land was considered for all the studied air masses. Similar to Sect. 4.1, for each of the back-trajectory points along their path, air masses were considered to be over land (or over the ocean) if at least 75% of the 125 trajectories in the set (see Sect. 2.2.2) were over land (or over the ocean) based on the land-sea mask from ECMWF ERA-5 reanalyses. The CS measured at Maïdo has not been included in this analysis since it can be an indicator of both sources and sinks for gaseous compounds, thus complicating the interpretation of the associated results. Figure 10 shows the correlations between the compound signals and the variables mentioned above (at 95% confidence level), and where applicable, the strength of these correlations; an extended version of Fig. 10, which also shows inter-species and inter-variable correlations, is reported in Fig. S16 of the supplement.

Considering first the correlations with the variables related to air mass history, sulfur containing compounds and iodic acid exhibit similar behaviour. They are in particular anti-correlated with the time spent by the air mass over the ocean since it last flew over land, with a particularly marked anti-correlation for MSA and $SO_5^-$ (R = -0.74 and R = -0. 75, respectively). However, there is, as expected, no indication of a  terrestrial source for these compounds (no correlation with time spent over the island or farther land).

We then observe that both oxalic acid and malonic acid signals (that were found correlated) are positively linked to the time spent by the air mass over land, particularly on land other than Reunion, and oxalic acid shows, in addition and similarly to the purely marine group, an anti-correlation with the time spent by the air mass over the ocean since the last land crossed. These observations suggest that these two compounds may have a remote terrestrial source that controls at least part of the observed variability (in addition to a possible marine source for oxalic acid). Concerning malonic acid, these findings contrast with the results of Peltola et al (2023), who measured overall higher signals in purely marine air masses than in land-influenced air from the Baring Head station in New Zealand. It should be noted that at Baring Head, land-influenced air masses are flying with significant time over New Zealand prior sampling, and potentially in the terrestrial BL where larger condensational sink favour the loss of marine species. Our FT observations also indicate that overflying land in the FT (only one of the air masses studied travelled in the BL over land other than Reunion) seems to be sufficient for these air masses to acquire a terrestrial signature via exchanges between the BL and the FT (although this does not affect the CS measured at Maïdo, see Sect. 4.1), and that they retain this signature during the rest of their voyage in the FT. This finding is consistent with the results obtained at puy de Dôme by Farah et al. (2018), who showed that even after 75h spent in the FT, the sampled aeorosols preserve specific properties of their air mass type. Maleic acid, in contrast to all the other compounds identified, shows a positive correlation with the time spent by the air mass over the ocean since it last flew over land, which would indicate a marine source at low wind speed.

Although the value of the correlations observed with variables relating to the history of the air mass (even though they are mathematically significant) may be questioned for amines given the supposedly short lifetime of theses species, the correlations obtained with the variables indicative of terrestrial influence (i.e. the number of hours since the last passage over land, the time spent over land and the time spent over Reunion for air masses that have never flown over other land masses) all seem to converge towards a stronger terrestrial signature of the air masses associated with the highest C2 amine signals. The fact that

there is a correlation with the time spent by the air mass over Reunion for amines suggests, on the one hand, that there may be a terrestrial source for these compounds, and the fact that this correlation is particularly marked (especially compared with those obtained with the other variables describing the history of the air mass) is also consistent with a more pronounced impact of the last few hours of travel of the air mass before sampling of these compounds, which lifetime is, again, assumed to be

short.

Finally, a common feature of the purely marine compounds, also shared with oxalic acid and C2 amines, is the anti-correlation with the mean radiation retrieved by ECMWF ERA-5 reanalyses along the air mass backtrajectory. The explanation for this anti-correlation, which is moderate for most of the compounds, is however uncertain; it may be that more intense radiation favours the photochemical processes that are a sink for these compounds.

If we now consider the correlations between the species signals and the variables concurrently measured at Maïdo (Fig. 10), a positive correlation with temperature is observed for many compounds (with the exception of C2 amines and maleic acid), moderate overall for sulfur-containing compounds (R<0.30), intermediate for iodic acid and malonic acid (R~0.47) and stronger for oxalic acid (R~0.68). The interpretation of the link with temperature is uncertain; however, one possible hypothesis is that an increase in temperature in the FT could result from stronger local intrusions from the (warmer) BL. With the exception

of maleic acid, all the identified compounds show in addition an anti-correlation with the RH measured at Maïdo, which is particularly marked for oxalic acid (R = -0.75) and clusters in the MSA-derived group (R = -0.71). There are several possible explanations for this observation. As suggested in the previous section for MSA, this could indicate the occurrence of an evaporation from the condensed phase (favoured at low RH). For oxalic acid, for example, which is usually the dominant dicarboxylic acid in the particulate phase with malonic acid (both in aerosols measured from the continental sites and over

open ocean waters; Kawamura et al., 2016), the correlations observed in the gas phase with temperature and RH are consistent with the findings of Clegg et al. (1996). In fact they showed that the occurrence of gaseous oxalic acid in the atmosphere should be promoted under conditions of combined low RH, low aerosol pH and increased temperatures (>15°C). The hypothesis of evaporation also appears to be an interesting explanation for the presence of C2 amines, but it would need to be explored in greater depth using dedicated measurements in the future. Indeed, despite the observed anti-correlation, the C2

amines signals measured for the highest RH are not the lowest (figure not shown); moreover, while Kürten et al. (2016) mention evaporation from the condensed phase to explain the variability of the C4 amines signal at Melpitz, it was neglected in the modelling study of Yu and Luo (2014), which may raise questions about the real importance of the evaporation process for amines. Apart from possible evaporation from the particulate phase, the anti-correlation of the signal of most of the species identified with RH is certainly indicative of a higher sink at high RH, either related to the presence of cloud (and subsequent

dissolution of gaseous compounds in the droplets) or because high RH may also favour chemical reactions that act as a sink for these compounds. Maleic acid signals, on the contrary to most species identified, vary positively with the time spent by the air mass in a cloudy environment which may indicate a liquid chemical source.

In the end, we were able to highlight a number of correlations between the signals of the compounds identified and the variables considered; however, although significant, these correlations often remain moderate, and sometimes difficult to interpret. The

results of this analysis must therefore be treated with caution, but they seem to point to a terrestrial origin for oxalic acid, malonic acid and amines (which may be more distant for oxalic acid and malonic acid compared to amines), which likely superimposes on a marine source for oxalic acid and amines. The other compounds (i.e. sulfur containing compounds, iodic acid and maleic acid) appear to have a dominant marine origin. These results are summarised in Table 2.

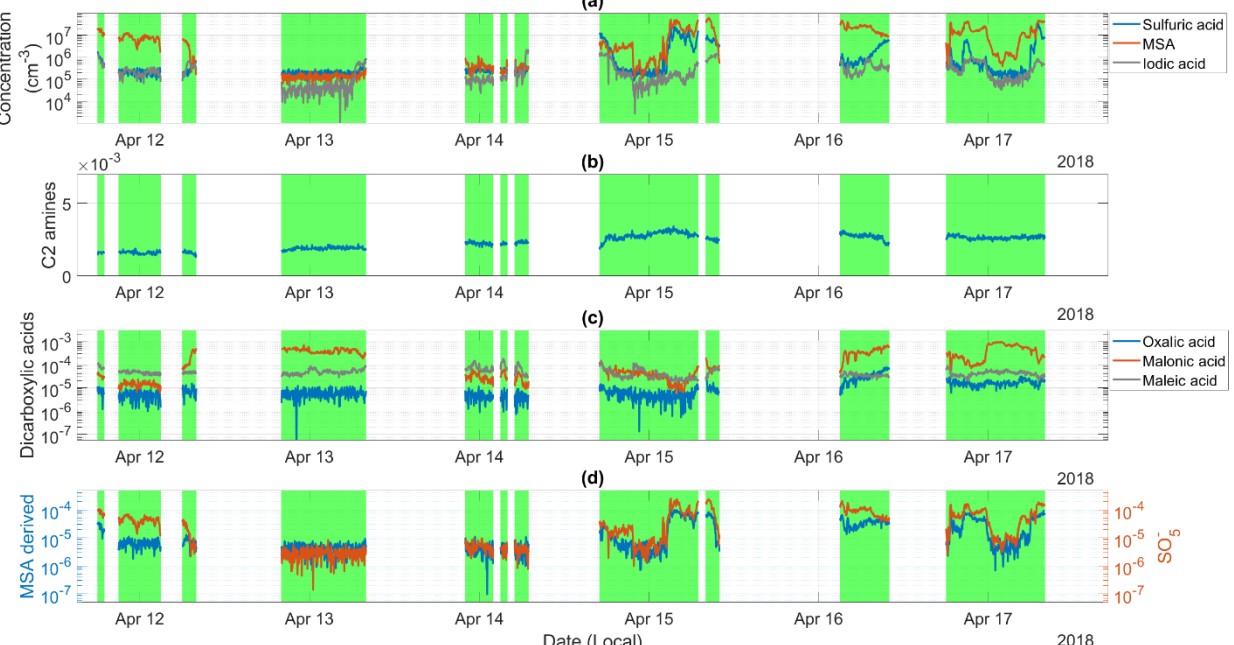

Figure 9 Same as Fig. 8 but with a focus on the periods during which the site is in the FT, to facilitate the visual inspection of signal variability in these specific conditions. Because of the suspected contamination affecting C4 amines, these are not shown in this figure.

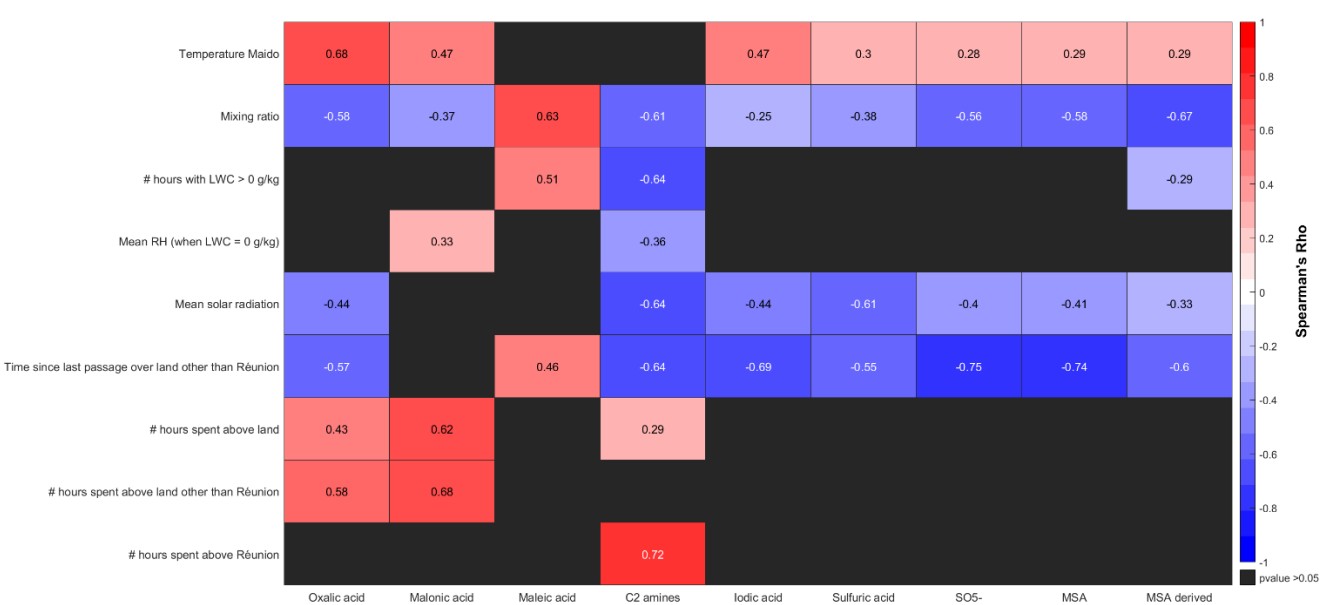

Figure 10 Spearman correlation coefficients between the signal of the species identified in the FT at Maïdo and a series of variables concurrently measured at the station (temperature and RH) or related to the air mass history (and retrieved in ECMWF ERA-5 reanalyses), including the mean radiation (calculated from values > 10Wm⁻²), the number of hours with non-zero liquid water content (LWC) along the air mass path, the mean RH calculated outside cloudy areas, the time spent by the air masse over the ocean since it last passed over land, the time spent over Reunion for the air masses having never flown over a land other than Reunion (during the 72h of back-trajectory considered) and for all the air masses the total time spent over land. The colour of each square indicates the strength of the correlation, with the square being black if the corresponding p value is above 0.05.

**Conclusions**

The present study contributes to documenting the composition of the MFT, which has been little explored until now although it is in particular supposed to be the privileged site of NPF in the marine atmosphere. Observations performed at the Maïdo high altitude station, which is ideally located in the Southern Indian Ocean, are presented to address this objective.

The data collected during the BIO-MAIDO campaign during March-April 2019 were first analysed in synergy with the predictions of the BL height provided by the mesoscale atmospheric model Meso-NH over the same period in order to identify a tracer of FT conditions at the site. The standard deviation of the wind direction ($\sigma_\theta$), which can easily be derived from continuous meteorological measurements at the station, was found to be a relevant parameter to distinguish periods when the site is under the influence of the BL (i.e. mainly during the day), from periods when the conditions at the station are more

representative of the FT (i.e. mainly during the night, under the influence of a strengthened large –scale subtropical subsidence). On the other hand, the outputs of Meso-CAT, which results from the coupling between the trajectory model CAT and the mesoscale atmospheric model Meso-NH (and which use is therefore subject to the availability of Meso-NH simulations), were used in a sensitivity study to evaluate the ability of CAT (which is based on ECMWF ERA-5 reanalysis and is therefore less resolved but can be run routinely) to assess in addition the positioning of air masses in the BL or in the

FT during the hours preceding their arrival at the site in the FT. The BL thicknesses retrieved by ECMWF ERA-5 reanalyses (and therefore CAT) do not seem reliable over the island, most likely due to the to the too coarse horizontal spatial resolution of the model. Meso-CAT indicates, however, that the time spent by the air masses over land before their arrival at the site in the FT is overall limited, and that it seems furthermore reasonable to assume that they are systematically located in the FT during the last hours of their journey. ECMWF ERA-5 reanalyses, on the other hand, are useful for tracing the synoptic origin

of air masses and can be used to assess the positioning of air masses in the FT or in the BL along their path over the ocean. The chemical composition of molecular clusters and potential precursor molecules present in the MFT was in a second step investigated using measurements performed with a nitrate based CI-APi-TOF deployed at Maïdo in April 2018, during the OCTAVE campaign. During the investigated 7-day period, FT conditions were (as expected) detected mainly at night at the site; the sampled air masses travelled mainly over the ocean, in the FT, and were therefore reasonably considered to be

representative of the MFT. A number of clusters and molecules were identified and classified into 9 groups according to their chemical composition: fluorinated species (likely originating from the sampling lines), iodic acid, sulfuric acid, $SO_5^-$, MSA, MSA-derived (including $CH_4O_3S \cdot HSO_4^-$ and $CH_4O_3S \cdot IO_3^-$), dicarboxylic acids (including oxalic acid, malonic acid, and maleic acid), C2 amines and C4 amines (with a suspected contamination). Most of these compounds have signals that tend to be higher in the daytime BL conditions, or are similar in the BL and in the FT; interestingly, MSA and C2 amines show signals

that are, in contrast, on average significantly higher in the FT. In order to get further insight into the factors contributing to explain the variability of the signals mesured in the FT, the correlations with a series of variables concurrently measured at the site or related to air mass history were investigated. Although the results of this analysis must be considered with caution (as most of the correlations remain moderate and some are not easy to interpret), they seem to indicate a terrestrial origin for oxalic acid, malonic acid and C2 amines, likely combined with a marine source for oxalic acid and amines. The findings for

iodic acid, sulfur species and maleic acid point to a dominant marine origin.

Because FT conditions are mainly identified at night and the available dataset was relatively short (7 days), the possible involvement of the observed species in (daytime) NPF was not addressed in this study. The presence of the identified compounds suggests, however, that there is a real NPF potential in the MFT as the involvement of many of these species in the different stages of daytime NPF has been reported in other environments. Notwithstanding that further work is needed to

fully characterize MFT NPF, including the acquisition of longer-term data sets in order to have a higher probability of obtaining daytime FT data, the reported dataset and associated results are are a highly valuable contribution to documenting the overall composition of the MFT, and indirectly to the understanding of the processes occurring in this specific region of the atmosphere.

## Data availability

The data shown in the figures can be found under https://doi.org/10.5281/zenodo.10302533 (Salignat et al., 2023).

## Author contributions

JB and MR contributed to the coordination of the OCTAVE 2018 campaign. CR, MPR, SI and JMM performed the measurements during the OCTAVE campaign. PT run the atmospheric mesoscale model Meso-NH, J-LB run the trajectory model Meso-CAT and made the trajectory CAT model available to be used by RS. RS, CR, MR, SI, KS analysed the data. RS and CR wrote the paper. All co-authors contributed to reviewing the paper.

## Competing interest

At least one of the (co-)authors is a member of the editorial board of Atmospheric Chemistry and Physics.

## Acknowledgements

The OCTAVE 2018 campaign was performed in the framework of the OCTAVE project of the "Belgian Research Action through Interdisciplinary Networks" 5 (BRAIN-be) research programme (2017–2021) through the Belgian Science Policy Office (BELSPO; contract no. BR/175/A2/OCTAVE). We would like to thank UAR 3365 of OSU-Reunion for its support with respect to the deployment of the instruments. The Maïdo site is operated with the support of Université de la Reunion, Météo-France, CNRS-INSU - under the long-term observation programme - and the French Ministry for Research - under the ACTRIS-FR national research infrastructure. The French SNO-CLAP programme is also acknowledged for supporting continuous aerosol measurements at the Maïdo observatory. Meso-NH simulations have been made on Météo-France supercomputer. The ERA5 reanalysis data used in this study were provided by the ECMWF (https://doi.org/10.24381/cds.adbb2d47 and https://doi.org/10.24381/cds.bd0915c6) and generated using Copernicus Climate Change Service information (C3S). We thank the colleagues of the Royal Belgian Institute for Space Aeronomy for providing us the meteorological data associated to the FTIR experiment at the Maido observatory.

## Financial support

Support was received from the European Union's Horizon 2020 Research and Innovation programme (ACTRIS TNA; grant agreement no. 654109) to organize part of the measurements conducted during the OCTAVE 2018 campaign. The BIO-MAÏDO project was funded by the Agence Nationale de la Recherche (ANR-18-CE0-0013-01). Matti Rissanen also appreciates funding from the Academy of Finland (project nos. 299574, 326948, 331207 and 346369). Karine Sellegri and Matti Rissanen have received funding from the European Research Council (ERC) under the European Union's Horizon 2020 research and innovation programme (Grant agreements No. 771369 and 101002728, respectively). The Sea2Cloud project is endorsed by SOLAS.

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
