# Peer review of "Measurement Report: Insights into the chemical composition and origin of molecular clusters and potential precursor molecules present in the free troposphere over the Southern Indian Ocean: observations from the Maïdo observatory (2150 m a.s.l., Reunion Island)"

_EGUsphere, 2023_

## Author Comment (AC1)

We thank Reviewer #1 for his comments and suggestions which contributed to improve our manuscript. Our answers are listed below.

**Major comments:**

Comment 1.a: How were instrument backgrounds ("zeros") determined, and what was their result? This potentially important aspect of the trace gas measurements is missing (or, if I missed it, insufficiently discussed). I got interested in particular due to the detection of fluorinated acids, quite consistently throughout the week, and amines, with a continuous substantial presence, especially of C4 amines (Fig. 7). Indeed, the fluorinated species are suspected to be due to some instrumental contamination (P18, 1st paragraph), which zero measurements might be able to confirm, and implying the importance of zeros potentially also for other species.

Regular blanks, i.e., zero measurements, are utmost important in field deployments to enable discriminating instrumental and sampling artefacts from the real signals originating from the studied environment. These are usually performed by feeding the instrument with a contaminant free gas, generally as clean as possible air or nitrogen. During the blank experiment the signals from the sampled ambient environment should go down to zero level, whereas instrumental and sampling related artefact signals should remain relatively constant. Unfortunately, blank measurements were not available during this campaign due to the complex experimental setup position.

In connection with Comment 15, we now clearly indicate the absence of blanks for all compounds (and not only amines), and we discuss more thoughtfully contamination issues related to fluorinated compounds and possibly amines in Sect. 4.2.1. It should also be noted that the comparison between the C2 amines signal measured at Maïdo and reported in the work of Brean et al. (2021) has been slightly updated following the correction made to the amines signal normalization mentioned in the reply to Comment 24 (which led to an increase in the normalized signal).

About fluorinated compounds:

*"As shown in Fig. 7.b, the fluorinated compounds highlighted in Fig. 6 constitute a non-negligible fraction of the identified signal, 18.2%, with no marked difference between BL and FT conditions (on average 18.3% and 17.9%, respectively). Since blanks were not performed during this campaign due to the complex experimental setup position, the origin of these compounds remains uncertain but, similar to Ehn et al. (2012), we suspect that these molecules are contaminants emitted by fluorinated plastic (i.e. polytetrafluoroethylene, PTFE, in this case), present in our case in the tubing used in the CI-inlet setup."*

About amines:

*"The C2 amine signals measured at Maïdo are higher than those reported by Brean et al. (2021) in the MBL (i.e. closer to the ocean, which is one of the identified sources of amines), but they show a similar diurnal cycle, with a minimum during daytime (Fig. 8.b). In contrast, while similar behaviour is observed by Brean et al. (2021) for C2 and C4 amines, there is no marked diurnal cycle in C4 amines signal at Maïdo (which constitute an even larger fraction of the identified signal, on average 28.6% and 27.2% measured in the FT and in the BL, respectively, Fig. 7.b), but often a clear peak when the air mass is at the interface between the BL and the FT (Fig. 8.b)."*

*"In absence of dedicated instrument calibration and blanks, the actual concentration associated with the amines signal measured at Maïdo could not be assessed, which unfortunately limits our analysis. In the absence of blanks, in particular, the risk of a contamination in the measurements cannot be completely excluded. It should be noted, however, that there is no correlation between the temperature and the intensity of the C2 amines signal, and that the C4 amines signal is anti-correlated with*

*temperature (Fig. 10). This therefore rules out the possibility that these compounds are the result of a temperature modulated contamination by the instrument or the sampling line (volatilization-related). We also ruled out the possibility of an interference between the signal of the cluster composed of $NO_3^-$ and 6 water molecules (mass 170.052 Th) and that attributed to the C2 amine cluster $(C_2H_7N)(HNO_3)_1 \cdot NO_3^-$ (170.042 Th) reported by Kürten et al. (2016) at high RH. Indeed, the time series of the signals associated with the two peaks in the C2 amines group show the same variability (figure not shown), and the signal attributed to the cluster with nitrate dimer is furthermore anti-correlated with RH (R = -0.18), and even more with the water mixing ratio (R = -0.42). On the other hand, we cannot exclude the presence of a constant background signal (for e.g. caused by the use of a compressor for the generation of the sheath air used in the CI unit), on which would be superimposed a "real" ambient signal determining the observed variations. In this case, however, the amplitude of the reported signals would obviously be overestimated (and as would be that of the associated fractions shown in Fig. 7), but the conclusions related to signal variability would be preserved. We therefore believe that despite the above-mentioned limitations and associated uncertainties, our observations are interesting and should call for further measurements of gaseous amines in the MFT as they may question the existence of an unidentified source."*

**Comment 1b**: The time series for the C4 amines additionally spiked my interest, as it exhibits numerous "spikes" followed by slow decays (Fig. 8). Are those instrumental or real? Either way, can the authors speculate on the causes for those spikes?

As indicated in the manuscript, these peaks are regularly observed at the interface between the BL and the FT (P21, L3-6) and would therefore result from the mixing of these two different air masses. There is no clear indication that the amines we observe are the result of a contamination (see reply to Comment 1.a); on the other hand, current knowledge of these compounds did not allow us to identify any obvious source which might, given the expected short lifetimes of these compounds, explain their presence in the FT. All these aspects are discussed in the manuscript (Sect. 4.2), which at once reviews current knowledge and observations, describes the limitations of the analysed dataset (all the more comprehensively now in connection with Comment 1.a) and indicates future research needs in relation to these aspects. We therefore feel that there is no need to speculate further on the observed variations.

**Comment 2**: Do the authors have idea about the identity of the other major peaks observed (prominent unidentified mass spectral peaks in Fig. 6a)?

They should at least discussed. Even if unidentifiable, a mass defect diagram could provide clues?

In order to adress the questions of the two Reviewers regarding the unidentified fraction of the signal, we have added a paragraph at the end of Sect. 4.2.1 which provides some additional elements:

*"As illustrated in Figs. 6 and 7 and mentioned above, a significant part of the signal measured with the CI-APi-TOF has not been identified, of which a certain fraction may be made up of organic compounds, and given their mass range in particular highly oxygenated molecules (HOMs). An attempt to identify the compounds observed in Hyytiälä, the birthplace of HOM studies located in the boreal forest (Ehn et al., 2012; 2014), was however unsuccessful. In particular, the molecules that were used as fingerprint for the study of HOM sources by Yan et al. (2016), are not detected at Maïdo (see Table S1 in the Supplement for a list of these compounds). A first explanation for the absence of these HOMs at Maïdo is certainly linked to the fact that their precursors, monoterpenes, are present at low concentrations, below instrumental detection limits during the night (Rocco et al., 2020). Additionally, the meteorological conditions (mainly temperature and relative humidity) and oxidant concentrations, which all certainly affect the chemical reactivity, are very different at Maïdo compared to Hyytiälä.*

*Figure 6 also highlights a certain number of peaks associated with high signals which have not been identified, in particular at UMR 114, 134, 177, 197, 246, 260, 310, 334 and 362 Th. Since these compounds do not seem to be enriched in the FT (Fig. 6.b), they have not been the subject of in-depth study, but the time series of their signals are nonetheless shown in Fig. S14. The signals associated with UMR 114, 134, 177 and 197 Th (Fig. S14. a) show similar variations. Given the difference of 63 Th (corresponding to HNO₃), masses 114 and 177 Th on the one hand, and 134 and 197 Th on the other, are most likely associated to the same compound, therefore explaining the similarities within each pair, and the similitude between the two pairs is certainly more broadly indicative of a common source. The other masses (UMR 246, 260, 310, 334 and 362 Th) have a different behaviour (Fig. S14.b), which is actually comparable to that observed for the peaks attributed to C4 amines (Figs. 8 and 9). These observations may be the subject of more in-depth investigations in future studies dedicated to the BL composition."*

[Figure]

Figure S14 Timeseries of the normalized signals of UMR a. 114, 134, 177 and 197 Th and b. 246, 260, 310, 344 and 362 Th. Green patches depict the hours when the Maïdo station is in the FT based on the analysis of the standard deviation of the horizontal wind direction.

**Comment 3**: "Molecular clusters" (title and text). The term "molecular clusters" in the title (plus corresponding throughout the text) is misleading. Measurements were primarily made of certain acids plus two classes of amines. The only observed clusters (less the reagent ions/clusters) were made up by up to 2 sulfuric acid molecules. Often the reference is to the observed "ion clusters", which are hence primarily a product of the measurement method but not as such present in the atmosphere. If that understanding is correct, I recommend adjusting the title (and text).

Among the identified signals, several have been attributed to actual clusters: sulfuric acid clusters but also clusters in the so-called MSA-derived group (including $CH_4O_3S \cdot HSO_4^-$ and $CH_4O_3S \cdot IO_3^-$). The other identified species are not, in fact, clusters, but rather molecules. It would seem, however, that the term "clusters", though misleading or even inexact, is used in the literature to designate both the clusters and molecules identified with similar technique, which is why we propose to use it in this study too. This is now clearly stated at the beginning of Sect. 4.2.1:

*"Note that, for the sake of simplicity and in line with accepted usage in the literature, we will hereafter use the term "cluster" to designate both "real" atmospheric clusters and clusters resulting from the chemical ionization process in the instrument, which in reality reflect the presence of neutral molecules."*

Concerning the use of the expression "ion clusters", it is indeed inappropriate in the context of this work; however, we have only found the reference to ions in the legend of the ordinate axis of Fig. 7, and have therefore replaced it by *"Fraction of identified compounds (%)"*, and in the header of Table 2, that we also modified.

**Comment 4**: Section 4.2.2 overall (and Figs. 10, S10):

   a: I wonder how significant are correlations with "time spent over land other than Reunion" and "time since last passage over land other than Reunion" are, as they could only be quantified for one of the nights, if I understand correctly?

As illustrated in Figs. 5 and S6 (now Fig. S9), air masses that have flown over land other than Reunion and arrive at Maïdo when the site is in the FT were sampled on several nights, and in particular on the nights of 12-13, 15-16 and 16-17. Altogether, out of the 65 hours during which the site was in the FT over the studied period, the air masses that reached the station travelled over land other than Reunion in 53% of the cases. This suggests, therefore, that the "time spent over land other than Reunion" and "time since last passage over land other than Reunion" are relevant to consider in the correlation analysis.

   b: Can the authors speculate about the mechanisms of "time spent over Reunion" affecting FT composition, in particular amine concentrations? I have not discerned that from this rather detailed and complex discussion of that analysis.

Indeed, the objective associated with the study of this specific parameter was not clearly stated in the manuscript. The goal is to identify a possible nearby terrestrial origin for the observed compounds, with the hypothesis that the longer the time spent over the island, the more likely it is that this source has impacted the observations through exchanges between the boundary layer and the free troposphere. This is now clearly indicated in Sect. 4.2.2:

*"It should be noted that the air masses having never flown over a land other than Reunion (during the 72h of back-trajectory considered) were not included in the correlation analysis with this last parameter (because the exact time since their last passage over land is unknown). The time spent over the island was analysed instead for these particular air masses to provide a possible additional indication on the variability of the associated signals, and identify in particular a possible nearby terrestrial origin for the observed compounds. The underlying assumption is that the longer the time spent by the air mass over the island, the more likely it is that a nearby terrestrial source may have impacted the observations at Maïdo through exchanges between the BL and the FT."*

   c: Maybe a table of hypothesized source mechanisms for the observations of the various compound groups could help in summarizing this section.

This is a very good suggestion; we have added a column to Table 2 to summarize the results of this analysis.

   d: I am not an expert with NO3-CIMS in particular, but could the anti-correlation of most signals with RH be connected to instrument sensitivity potentially varying with RH?

Nitrate ion charging is generally insensitive to water (Rissanen et al., 2014; Hyttinen et al., 2017), and thus it seems unlikely that the observed anti-correlation would be due to changing RH.

**General comments:**

**Comment 5**: Up to Section 3.2, the impression is given that a new method is found of establishing FT vs BL conditions via a tracer (e.g., P3 L26 "paper presents a tracer", P4 L9 "identify a tracer", P5 L34 "determine a tracer"). Or I have found those formulations misleading. Only in 3.2, it becomes apparent (to me) that the same approach as in Rose et al. (2017) and other papers is used (including preceding ones cited in Rose et al.), except, I believe, for a weaker threshold value.

I would consider making that clearer already earlier.

Indeed, the purpose of this work was not to develop a new approach for identifying FT conditions, but rather to identify which of the methods existing in the literature would allow an easy and efficient classification of the air masses sampled at Maïdo. In order to make that clearer already in the introduction we have modified the sentence P3 L26 to:

"*The first part of this study is dedicated to investigating the suitability of existing methods to detect in a simple way, from the continuous measurements performed at the site, the periods during which the station is in the FT (Sect. 3.1).*"

**Comment 6**: A figure illustrating the various models' horizontal grid sizes and locations would be useful (including a length scale). E.g., based on the map shown in Fig. S5 (which misses the scale by the way).

Given the variety of resolutions associated with each of the simulation/model domains, it is complex to represent all the grids on a synthetic figure. However, in order to clarify at least the positioning and size of the domains used by Meso-NH, we have represented them on a figure that was added to the supplement (Fig. S1). We have also added the scale on Fig. S5 (now Fig. S6).

[Figure]

Figure S1 The extent of Meso-NH's three domains with horizontal resolution of 2000 (d01), 500 (d02) and 100 m (d03).

**Minor comments:**

**Comment 7**: Abstract:

I would consider restructuring the abstract so that the identification of FT conditions at the site (plus implications) is described BEFORE the findings regarding cluster compositions and sources. I.e., first describe the setting, then the results. (Like in the text actually.)

We do not think it is essential that the progression of the abstract should be identical to that of the paper. We have chosen to present first in the abstract the results that make this paper innovative, and then to mention the method used to identify the FT conditions which are at the heart of the study.

**Comment 8**: P1 L23: would remove the latter "the", as I am suspecting that not all clusters were detected. [Later edit: see also comment (3) above regarding "clusters".]

Done.

**Comment 9**: P1 L25-26: I believe this sentence has some grammar issue or is missing a part.

We have split the sentence as follows to facilitate its understanding:

*"A number of clusters were identified and classified into 9 groups according to their chemical composition; among the identified species, the groups containing methanesulfonic acid (MSA) and C2 amines show signals that are on average significantly higher when the site is under conditions representative of the marine FT (compared to the BL)"*.

**Comment 10**: P2 L13: References to two technical papers are given, but missing somewhere here are references to studies that have reported on molecular cluster/precursor observations related to NPF.

Although it is addressed in Sect. 4.2, this is an aspect that is indeed missing in the introduction. We now address it also in the introduction, with appropriate references. In response to Comment 11, we have also included information related to the definition of the planetary boundary layer and the free troposphere. The manuscript has been modified / completed as follows:

*"NPF has been extensively studied during the last decades, and reported to occur in almost all known environments (Kerminen et al., 2018). Although there are still gaps in our knowledge, the development of mass spectrometry techniques achieved at the same time has improved our understanding of the mechanisms by giving access to the chemical composition of the molecular clusters and their precursors (Junninen et al., 2010; Jokinen et al., 2012). Besides sulfuric acid, which is commonly accepted as a key driver of NPF, the nucleating ability of other compounds has been demonstrated, both from observations in the real atmosphere and experiments in simulation chambers. The role of ammonia (Kirkby et al., 2011) and amines (Almeida et al., 2013) in stabilizing sulfuric acid clusters has in particular been highlighted, and the potential of organic compounds of biogenic origin to also contribute (Schobesberger et al., 2013; Riccobono et al., 2014; Lehtipalo et al., 2018) and even nucleate on their own (Kirkby et al., 2016; Rose et al., 2018) has been reported. In the marine environment, although observations are limited, the role of iodic acid has been more specifically identified in coastal areas (Sipilä et al., 2016; Baccarini et al., 2020; Beck et al., 2021; Peltola et al., 2023). However, most observations in the real atmosphere have been made in the planetary boundary layer (BL), i.e. in the lower part of the troposphere that is directly influenced by the earth's surface (Stull, 1988). Observations are in contrast more scarce in the free troposphere (FT), i.e. at higher tropospheric altitudes above the BL, mainly due to the difficulty of sampling this remote region of the atmosphere."*

**Comment 11**: P2 L14 (or earlier): It would be instructive to the reader to briefly explain the difference between BL and FT. "Remote region of the atmosphere" could also refer to near-surface air except in remote areas, such as the marine BL. The key is of course the vertical layering of the atmosphere and its relation to vertical mixing. I am missing here any reference to the vertical. (The FT is, in a way, always quite close: just a few kilometers (or less) up.)

As indicated in the response to the previous comment, we have included indications of the vertical distribution of the boundary layer and the free troposphere.

**Comment 12**: P2 L25: Which "these events" is being referred to?

These events refer to MFT NPF events. The sentence has been slightly modified to facilitate its understanding:

*"There is, however, no direct evidence of the implication of sulfuric acid in the process since these MFT NPF events have not been the subject of any detailed chemical characterization."*

**Comment 13**: P3 L22: How far inland is the Maïdo station?

The Maïdo station is about 15km inland as the crow flies. This information has been added to the manuscript (Sect. 2.1.1):

*"The measurements used in the present work were performed at the high-altitude observatory of Maïdo, which is located on Reunion Island, about 15 km inland from the west coast (21.080∘ S, 55.383∘ E; 2150 m a.s.l.)."*

**Comment 14**: P4 L~20: Was there any humidity control part of the inlet/DMPS setup?

The sample is indeed dried before entering the DMPS, so that its relative humidity is kept below 40%, in line with the recommendations of Wiedensohler et al. (2012). This is now clearly stated in the manuscript:

*"The DMPS was operated behind a whole air inlet which is characterised by a higher size cut-off of 25 µm for an average wind speed of 4 m s⁻¹. The air is dried before entering the instrument, so that the relative humidity of the sample is kept below 40%, in line with the recommendations of Wiedensohler et al. (2012). DMPS measurements were used in the calculation of the condensation sink (CS)…"*

**Comment 15**: P5 L7: A lack of "blanks" for amines is noted. How? Wouldn't any blank always blank all compounds at the same time, as a TOF-MS was used?

The wording is indeed confusing, as the blank does in fact concern all the detected compounds. As previously indicated in the response to Comment 1, we have therefore modified the sentence in Sect. 2.1.2, and indicated the absence of blanks later in Sect. 4.2.1:

- Sect. 2.1.2: *"For amines, in contrast, in the same way as Brean et al. (2021), we only report ion signals since we did not perform specific in situ calibration for these compounds."*
- Sect. 4.2.1: *"Since blanks were not performed during this campaign due to the complex experimental setup position, the origin of these compounds remains uncertain but, similar to Ehn et al. (2012), we suspect that these molecules are contaminants emitted by fluorinated plastic (i.e. polytetrafluoroethylene, PTFE, in this case), present in our case in the tubing used in the CI-inlet setup."*
- Sect. 4.2.1: *"In absence of dedicated instrument calibration and blanks, the actual concentration associated with the amines signals measured at Maïdo could not be assessed, which unfortunately limits our analysis. In the absence of blanks, in particular, the risk of a contamination in the measurements cannot be completely excluded. …"*

**Comment 16**: P5 L10-15: I wonder if the detailed explanation of how amine signals were normalized in various figures wouldn't be better placed later or in the figure captions.

The "specific treatment" applied to the amines signal in Figs. 8 and 9 is already recalled in the captions of these figures.

**Comment 17**: Table 1: Suggest to clarify somehow, which rows are for instruments and which for models.

Table 1 has been modified to follow the suggestion of the Reviewer.

**Comment 18**: P7, section 3.2., 1st paragraph:

The first paragraph talks about previous works studying BL development on Reunion but is very vague on what those studies have actually found ("diurnal cycle", "result of complex combination..."). I understand that this study's approach is conceptually simpler, but as the previous works are brought up, it would be more useful to the reader to briefly explain how the BL has been found to generally develop over the mountainous terrain around Maïdo.

We have followed the Reviewer's recommendation and added some elements from the study by Lesouëf et al. (2013) explaining the development of the BL over the Maïdo relief, together with references to more recent studies on the topic. In order to keep the message of Sect. 3.2 focused on

identifying the most appropriate tracer for distinguishing FT conditions at Maïdo, we have integrated this information into the section describing atmospheric dynamics around Reunion (currently Sect. 3.1). Following the recommendation of Reviewer #2, this section has been moved to Sect. 2 (new Sect. 2.1.1).

*"The pioneering case study of Lesouëf et al. (2013) has allowed, based on a model-measurement synergy, to get particular insight into the development of the BL on the slopes of Maïdo. Detailed characterisation of the early morning (07:00 – 08:00 LT) circulation over the mountain's western slope reveals the presence of two atmospheric layers on the chosen day (November 26th, 2008). Moist marine air is advected in the lower layer (~1600 m thick) as a result of wake vortices in the lee of the island, while the upper layer corresponds to FT air driven by the easterly trade winds. In the following hours, the model indicates that the circulation near the surface is dominated by upward flow which brings humid air to the station's altitude. As a result, the observatory lies under the influence of low-level air between mid-morning and early evening, before trade winds evacuate humid air and replaces it with FT air once upward transport has ceased. Studies conducted since the work of Lesouëf et al. (2013) have further highlighted the complexity of the regime that actually impacts Maïdo during the day (Duflot et al., 2019; Leriche et al., 2023; El Gadchi et al., submitted). The station lies at the confluence of thermal breezes from the west and the easterly trade winds, and it is the interaction between the trade winds and the island's relief which modulates the influence of these two flows on the site."*

**Comment 19**: Section 3.3:

In the beginning, it is pointed out that Meso-NH simulations were not performed for one period (OCTAVE), so that use of the CAT model was evaluated for the case of another period (BIO-MAIDO). I don't see the logic here right away? And then the 2nd paragraph continues by talking about Meso-NH and ECMWF simulations, causing more confusion. I assume ECMWF means CAT and onward discussion (incl. Fig. 3) is for the BIO-MAIDO period?

I suggest rewording and restructuring the story somewhat for a more consistent flow.

[Edit: I didn't remember Table 1 etc. when I read Section 3.3, explaining most of my confusion probably. Some more detailed or streamlines explanations here could be useful anyway.]

Indeed, as detailed in Sect. 2.2.2, CAT is a larger spatial scale model based on ECMWF wind fields while Meso-CAT is at a finer spatio-temporal scale, based on Meso-NH. The aim of Sect. 3.3 (now 3.2) is to find out whether, in the absence of Meso-CAT during OCTAVE, CAT can be used to study the history of air masses reaching the Maïdo, and in particular their position in the FT / BL. To do this, we "compare" the performance / results of the two models (CAT and Meso-CAT) during BIO-MAIDO, when both were available. The beginning of Sect. 3.3 (now 3.2) has been slightly reworded to recall/clarify these aspects in relation to the information provided in the previous sections:

*"…that may have influenced the composition of these air masses. As indicated in Table 1, Meso-NH (and therefore Meso-CAT) simulations have not been performed during the OCTAVE period. The possibility of studying the history of air masses arriving in the FT at the site with the less resolved CAT model (which uses ECMWF ERA-5 reanalyses, see Sect. 2.2.2) was therefore evaluated based on the simulations performed in the framework of BIO-MAIDO, when both Meso-CAT and CAT simulations were available (Table 1)."*

**Comment 20**: Fig. 5: I would add some labels to the maps (can be abbreviated, with explanations in the caption) to accommodate readers with less geographic background knowledge, at least for Africa, Madagascar, and the Ocean(s).

We thank the Reviewer for the suggestion; Fig. 5 as well as S6 (now Fig. S9) (and their captions) were modified accordingly.

[Figure]

Figure 5 72-hour back-trajectories of the air masses arriving at Maïdo (marked by the black star on the maps) when the station was in FT conditions during the nights of April a. 11 to 12, b. 12 to 13 and c. 14 to 15, 2018. The colour of each grid cell (0.2×0.2°) on the maps indicates the number of back-trajectory points falling into its area. Note that for each back-trajectory computed with the CAT model, all 125 trajectories of the corresponding set are shown in the figure. The abbreviations AF, MDG and IO stand for Africa, Madagascar, and Indian Ocean, respectively.

[Figure]

Figure S9 72-hour back-trajectories of the air masses arriving at Maïdo (marked by the black star on the maps) when the station was in FT conditions during the nights of April a. 13 to 14, b. 15 to 16 and c. 16 to 17, 2018. The colour of each grid cell (0.2x0.2°) on the maps indicates the number of back-trajectory points falling into its area. Note that for each back-trajectory computed with the CAT model, all 125 trajectories of the corresponding set are shown in the figure. The abbreviations AF, MDG and IO stand for Africa, Madagascar, and Indian Ocean, respectively.

**Comment 21**: P18 L31 - P19 L1: Please de-convolute this sentences. I cannot get my heard around it.

The sentence was split / modified as follows:

*"As in the case of sulfuric acid, there is a diurnal variation in the amplitude of the signals of the observed products in the iodic acid group, with on average higher values during daytime (Fig. 6.b) which logically translate into higher iodic acid concentrations in the BL than in the FT (Fig. 8.a, on average $7.19 \times 10^5$ $cm^{-3}$ and $2.90 \times 10^5$ $cm^{-3}$, respectively). This also leads to a contribution to the total signal identified which is higher in the BL than in the FT (Fig. 7.b, on average 2.5% and 1.0%, respectively), although significantly lower than that of the sulfuric acid group."*

**Comment 22**: P19 L2-4: Are these increases at sunrise/sunset significant? If so, it appears that H2SO4 exhibits the same behavior, at least on April 14?

We are not quite sure what the Reviewer means by significant. It is true, however, that sulfuric acid exhibits the same behaviour as iodic acid on April 14, and indeed this is also the case for MSA (Fig. 8.a), $SO_5^-$ and compounds in the MSA-derived group (Fig. 8.d). This suggests that it is the conditions observed

at the site on that day that may have simultaneously impacted all the species in these groups, and that the particular behaviour described for iodic acid (and observed at other sites) in fact only partially explains the variations on the 14th. The study of this specific event, which would require a more detailed analysis of the daytime BL conditions observed on that day, is beyond the scope of this paper. We have, however, completed the manuscript with the elements mentioned here:

*"… and is suspected by Baccarini et al. (2021) to be related to lower iodic acid formation yield at higher solar irradiance. However, since similar variations were observed for sulfur compounds on April 14, it may be that specific conditions at the site simultaneously influenced the variations of all these species on that day; deeper investigation of this event observed in BL conditions is nevertheless beyond the scope of this work."*

**Comment 23**: P19 L28-29: Would be nice to see that model fit in a supplemental figure.

As suggested, a figure illustrating the close match between measured MSA signals and the results of the linear combination including RH and CS in the FT is now included in the supplement (Fig. S12).

[Figure]

Figure S12 Scatter plot illustrating the close match between the results of the linear regression model determined from fitting RH and CS data to MSA concentrations measured in the FT (MSA = -5.63 x $10^5$ *RH + -1.97 x $10^{10}$ * CS + 5.20 x $10^7$, referred to as *Modeled MSA*) and measured MSA concentrations.

**Comment 24**: Figure 8:

What were signals normalized to in panels 2-4? Would be good to explicitly explain in the caption. For example, malonic acid fractions apparently exceed a factor of 10 at times. Does that mean its signal was 10 times that of the sum of the reagent ions (cf. P5 L1)? If so, I will have more comments, but probably there's a misunderstanding.

As indicated in the legend, the amines signal shown in panel (b) had been normalized by the nitrate dimer signal. We have corrected this normalization (also in Fig. 9) by taking the trimer signal instead, which is the approach actually recommended by Simon et al. (2016). This correction did not result in any change in the variability of the normalized signals, only in their amplitude, and the comparison with the observations of Brean et al. (2021) was corrected accordingly:

*"The C2 amine signals measured at Maïdo are higher than those reported by Brean et al. (2021) in the MBL (i.e. closer to the ocean, which is one of the identified sources of amines), but they show a similar diurnal cycle, with a minimum during daytime (Fig. 8.b)."*

For the compounds shown in panels (c) and (d), there is indeed an error, since despite what is indicated in the legend, these signals have not been normalized (which explains the abnormally high values noted by the Reviewer). We thank the Reviewer for noting this error, which we have corrected (also in Fig. 9); the signals in panels (c) and (d) are now normalized by the sum of the reagent ion signals.

[Figure]

Fig. 8

[Figure]

Fig. 9

**Comment 25**: P21 L17-21: Please de-convolute this sentence. I cannot follow.

The sentence was split as follows:

*"Given the expected limited lifetime and concentrations of gaseous amines, amine-driven nucleation is not expected to be efficient at high altitudes. On the other hand, the contribution of the process to CCN concentrations should also be minimal in the vicinity of highest amine sources, which are found in polluted areas and are therefore collocated with elevated CS (which tend to suppress nucleation). For these reasons amine-driven nucleation is not included in the global model of aerosol formation developed by Dunne et al. (2016)."*

**Comment 26**: P21 L24: "... than those measured" where and by whom?

The sentence has been slightly modified to facilitate its understanding:

*"the concentrations simulated by Yu and Luo (2014) were in particular found to be often significantly lower than the measured values to which they are compared in this same study."*

Also, we noticed that the paper by Yu and Luo (2014) was missing from the reference list, so we added it.

[revised manuscript text omitted]

---

## Author Comment (AC2)

We thank Reviewer #2 for his comments and suggestions which contributed to improve our manuscript. Our answers are listed below. Note that in addition to the points raised by the two reviewers, we have corrected the normalization of the amines signal in Figs. 8 and 9; in line with the actual recommendations of Simon et al. (2016), we now use the nitrate trimer signal instead of that of the dimer. This correction did not result in any change in the variability of the normalized signals, only in their amplitude, and the comparison with the observations of Brean et al. (2021) was corrected accordingly in Sect. 4.2.1:

*"The C2 amine signals measured at Maïdo are higher than those reported by Brean et al. (2021) in the MBL (i.e. closer to the ocean, which is one of the identified sources of amines), but they show a similar diurnal cycle, with a minimum during daytime (Fig. 8.b)."*

**Specific:**

**Comment 1**: Page 1 Title. The reviewer felt the title was only reflecting/summarizing part of the results presented in this study. Maybe adding the origin of air mass or clusters as well?

We have added the study of clusters' origin to the title, since it is indeed an important aspect of the analysis:

*"Insights into the chemical composition and origin of molecular clusters present in the free troposphere over the Southern Indian Ocean: observations from the Maïdo observatory (2150 m a.s.l., Reunion Island)"*

**Comment 2**: Page 3 Line 6. It seems a little bit contradictory to the sentences in Page 2 Line 15 where the authors cited several NPF observation in low FT or the interface of BL and FT with some of them from stationary measurements.

On Page 2 Line 15, we refer to observations in the low FT or at the interface between the BL and the FT irrespective of the terrestrial or marine environment associated with these measurements, whereas on Page 3 Line 6, we refer specifically to the marine FT, which has up to now been mainly sampled by aircraft.

**Comment 3**: Page 7 Section 3.1. The reviewer was wondering whether this section would fit better at the beginning of Section 2 as a separate subsection, considering it's mostly description of the measurement site from literature instead of results from this study?

This is indeed a good idea, so we have restructured Sect. 2 as follows:

*2.1 Observations*

*2.1.1 Atmospheric dynamics in Reunion*

*The measurements used in the present work were performed at the high-altitude observatory of Maïdo, which is located on Reunion Island, about 15 km inland from the west coast (21.080◦ S, 55.383◦ E; 2150 m a.s.l.). Reunion Island is located in the Indian Ocean, in the descending part of the southern Hadley cell… [Current Sect. 3.1]*

*2.1.2 Measurement site and instrumentation*

*The data sets obtained in the framework of two campaigns conducted at Maïdo were used in particular in the present work. Data collected during the BIO-MAIDO campaign (Leriche et al., 2023; https://anr.fr/Project-ANR-18-CE01-0013, last access: April 7, 2023) between March 14 and April 8 2019 were first used… [Current Sect. 2.1]*

**Comment 4**: Page 8 Line 9. It would be nice to add the frequency/fraction of BL thicknesses below 6m as FT conditions. Are there any periods at night that are still under BL conditions, or the station is always

at FT conditions at night? Based on Fig 4 from OCTAVE campaign, it doesn't seem to be always at FT at night.

We are not sure we understood the first part of the comment. A BL thickness of less than 6m is the criterion we have chosen to identify the periods when the station lays in the FT from the model outputs. On the basis of this definition, does the Reviewer want to know the fraction of time during which the station is in the FT according to the model? If so, according to the simulations performed with Meso-NH as part of BIO-MAIDO, the station was in free troposphere 55% of the time on the 26 days considered. This frequency has been added to the text, and for consistency, the frequency of FT conditions during OCTAVE (this time derived from the analysis of $\sigma_\theta$) was also added to Sect. 4.1:

- Sect. 3.1: *"In practice, for the rest of the analysis, BL thicknesses below 6 m predicted by the model were associated with FT conditions at the site (i.e. 55% of the time on the 26 days considered during BIO-MAIDO), and the rest of the points with BL conditions."*
- Sect. 4.1: *"Based on the analysis of $\sigma_\theta$, the station was found to be 36% of the time in the FT during the 7 days of interest in OCTAVE."*

In fact, as shown in Fig. 4, there are periods at night that are under BL conditions, which clearly demonstrates the interest of having a tracer other than an average time window for the distinction between FT and BL conditions. Nevertheless, Fig. S3 (now Fig. S4) shows that the station is mainly in the free troposphere at night (more than 80% of the time at night). These aspects are discussed in the last paragraph of Sect. 3.1.

**Comment 5**: Page 8 Line 15. The reviewer was wondering about the vertical wind. Would it be a better tracer to reflect the influence of BL at the site? Is the usage of horizontal wind instead of vertical wind due to the commonly-unavailability of vertical wind dataset? Could the authors clarify this a bit?

In fact, vertical wind should be a better tracer of vertical turbulence, but indeed, this variable seems to be less commonly measured than horizontal wind (it is at least not measured at Maïdo). This is now clearly mentioned in the manuscript:

*"It should be noted that vertical wind is obviously expected to be a better tracer of vertical turbulence, and therefore of FT and BL conditions, but this variable is not measured at Maïdo."*

**Comment 6**: Page 9 Line 15-16. It would be nice to provide a plot at least in SI to support the statement that the stricter threshold would not improve the results.

We are not sure to understand which type of figure is expected by the Reviewer, and we believe that the requested information is in fact already provided by Fig. 2. Indeed, in Fig. 2, we can already see that if we use a lower threshold / stricter criterion on $\sigma_\theta$ for the identification of FT conditions, the probability that the site is in the free troposphere is not significantly greater, unless we use a much lower threshold. For example, if the threshold on $\sigma_\theta$ was significantly lowered to 10°, the probability that the site is in the FT would become higher than 90%, but at the same time a higher number of FT observations would be excluded, which we want to avoid (we want to keep a balance between a sufficiently high probability of being in the FT and a criterion that on the other hand does not exclude too much FT data).

**Comment 7**: Page 12 Line 15-17. It seems bigger differences between the two models for Fig 3c-e compared to Fig 3b from northwest. Is there any explanation for this?

The idea here was to illustrate that the agreement between Meso-NH and ECMWF ERA-5 reanalyses concerning the estimation of the boundary layer thickness is overall much better over the ocean than over land. A detailed study of the variability of model performance over the ocean is beyond the scope of this work, and it is likely that the number of randomly selected points (4) is too limited for this

anyway (a statistical approach would be needed). However, in order to answer the Reviewer's question, we have calculated, for each of the 4 selected points, the average differences between Meso-NH and ECMWF ERA-5 reanalyses. The results are as follows:

Northwest: +137 / -175 m
Northeast: +235 / -100 m
Southwest: +203 / -131 m
Southeast: +194 / -76 m

For the point located to the north-west of the island, there are more periods during which the boundary layer thickness estimated by Meso-NH is lower (and more significantly) than that derived from ECMWF; as a result, the average of the negative differences (Meso-NH - ECMWF) is logically higher overall for this point, and the average of positive differences lower overall. Beyond the amplitude of the differences, however, it would appear that the variations in BL thickness calculated by the two models are more similar for the other points.

**Comment 8**: Page 13 Figure 4. There are FT periods with relatively high RH values (e.g. night of April 11-12, 13-14, 14-15). Would the water mixing ratios provide better comparison for FT vs BL differences?

We thank the Reviewer for his suggestion. The water mixing ratio (WMR) does indeed show overall more pronounced contrast between the FT and the BL than RH. However, since the variations in these two variables are closely correlated (see new Fig. S8 below), the conclusions drawn from the observation of RH remain valid when considering WMR. We have nonetheless added a figure showing the timeseries of the WMR to the supplement (Fig. S8), and completed the text in Sect. 4.1 as follows:

*"… although for temperature there is a bias due to the diurnal variation of the global radiation. As illustrated in Fig. S8, a more pronounced contrast between FT and BL conditions is observed when considering the water mixing ratio instead of RH (7.4 g kg$^{-1}$ (4.3 - 10.5 g kg$^{-1}$) in the FT vs 8.6 g kg$^{-1}$ (4.5-11.3 g kg$^{-1}$) in the BL), although the variations of these two variables logically appear to be strongly correlated. For simplicity, we have chosen to report only the analyses and results involving RH (which is measured directly) in the rest of the study, but it is worth noticing that the associated conclusions were confirmed when considering the water mixing ratio instead."*

[Figure]

Fig. S8 Timeseries of the water mixing ratio (blue) and relative humidity (RH, orange) during the OCTAVE campaign. Green patches depict the hours when the Maïdo station is in the FT based on the analysis of the standard deviation of the horizontal wind direction.

**Comment 9**: Page 14 Line 13-15. Consistent with the relatively high RH values of the night of April 11-12, 13-14, 14-15 in the reviewer's previous comment, it seems these nights have air masses from pristine marine air from Indian Ocean or southern Madagascar. The reviewer would probably category Fig S6a (night of April 13-14) to group 3 like Fig 5c (night of April 14-15) instead of group 2 considering the similarity (also shown in Fig S7). Is this because it's the night with air masses passing through terrestrial boundary layer other than Reunion as mentioned in Line 27-28? Could the authors clarify this a bit?

The situation observed on the night of the 13th to the 14th can be considered as a sort of intermediary/mixture between the situations illustrated in Figs. 5.b (group 2) and 5.c (group 3); the reason why we decided to associate this night with group 2 rather than group 3 is that some of the air masses sampled during that night flew over Madagascar, which is not the case for the air masses in group c, which remain over the ocean. This classification remains indicative, however, and is only intended to provide a synthetic illustration of the different situations encountered (with respect to horizontal spatial distribution of air mass back trajectories) during the period of interest; the classes defined in Fig. 5 are not used in the rest of the analysis.

**Comment 10**: Page 15 Line 17-21. Considering the much higher signals of reagent ions and isotopes have been excluded from the calculation of the total signal, what's the unidentified species (>75% fractions) over the mass range 80-400Th? The fraction seems a bit high. Are they mostly organics? They are labeled as "Others" in Fig 7a with pretty high signals. More description and discussion on this are needed.

There is indeed a significant fraction of the signal measured by the CI-APi-TOF that has not been identified in this work. In response to the questions from the two Reviewers about these compounds, we have added a paragraph at the end of Sect. 4.2.1 which provides some additional elements:

*"As illustrated in Figs. 6 and 7 and mentioned above, a significant part of the signal measured with the CI-APi-TOF has not been identified, of which a certain fraction may be made up of organic compounds, and given their mass range in particular highly oxygenated molecules (HOMs). An attempt to identify the compounds observed in Hyytiälä, the birthplace of HOM studies located in the boreal forest (Ehn et al., 2012; 2014), was however unsuccessful. In particular, the molecules that were used as fingerprint for the study of HOM sources by Yan et al. (2016), are not detected at Maïdo (see Table S1 in the Supplement for a list of these compounds). A first explanation for the absence of these HOMs at Maïdo is certainly linked to the fact that their precursors, monoterpenes, are present at low concentrations, below instrumental detection limits during the night (Rocco et al., 2020). Additionally, the meteorological conditions (mainly temperature and relative humidity) and oxidant concentrations, which all certainly affect the chemical reactivity, are very different at Maïdo compared to Hyytiälä. Figure 6 also highlights a certain number of peaks associated with high signals which have not been identified, in particular at UMR 114, 134, 177, 197, 246, 260, 310, 334 and 362 Th. Since these compounds do not seem to be enriched in the FT (Fig. 6.b), they have not been the subject of in-depth study, but the time series of their signals are nonetheless shown in Fig. S14. The signals associated with UMR 114, 134, 177 and 197 Th (Fig. S14. a) show similar variations. Given the difference of 63 Th (corresponding to $HNO_3$), masses 114 and 177 Th on the one hand, and 134 and 197 Th on the other, are most likely associated to the same compound, therefore explaining the similarities within each pair, and the similitude between the two pairs is certainly more broadly indicative of a common source. The other masses (UMR 246, 260, 310, 334 and 362 Th) have a different behaviour (Fig. S14.b), which is actually comparable to that observed for the peaks attributed to C4 amines (Figs. 8 and 9). These observations may be the subject of more in-depth investigations in future studies dedicated to the BL composition."*

[Figure]

Figure S14 Timeseries of the normalized signals of UMR a. 114, 134, 177 and 197 Th and b. 246, 260, 310, 344 and 362 Th.  Green patches depict the hours when the Maïdo station is in the FT based on the analysis of the standard deviation of the horizontal wind direction.

**Comment 11**: Page 18 Line 18-21. -Could the coinciding drop of RH be related to the advection of air masses from the higher altitude of the station? Can the models help to explain this?

As shown in the analysis at the end of Sect. 3.3 (Fig. S4, now Fig. S5), the majority of air masses reaching the site in FT conditions come from higher altitudes. In the absence of fine-scale simulations (performed with Meso-CAT) during OCTAVE, it is difficult to link RH drops to any particular behaviour of the air masses sampled at that time. We did, however, analyse the altitude of the air masses sampled when the site was in the FT during the night of the 14th to the 15th, based on the information provided by CAT (Fig. R1). While the altitudes of the air masses sampled between 7 and 9 p.m. were close to the site's altitude during the last 15-20 hours of flight before reaching the station, the altitudes were generally higher for the air masses sampled during the following hours, but without any apparent particularity for the time during which the RH drop was observed.

[Figure]

Fig. R1 Flying altitudes of air masses sampled at Maïdo during the hours when the site was in the FT during the night of April 14-15. Note that for each back-trajectory computed with the CAT model, all 125 trajectories of the corresponding set are shown in the figure.

**Comment 12**: Could the increase of SA be due to similar reason as the increase of MSA, i.e., the evaporation from condensed phase (Page 19 Line 26)?

We do not disagree with the possibility that evaporation from the condensed phase, possibly from evaporating cloud droplets, could be a potential source of gas-phase sulfuric acid. However, to the best of our knowledge, there is no literature that explains the underlying process. We do want to note that sulfuric acid is much less volatile than MSA, and consequently has much lower evaporation rates from gas-phase clusters than MSA (Rasmussen et al., 2022). This leads us to believe that the processes involved in the evaporation from the condensed phase of MSA and sulfuric acid are different. However, given that our work is primarily a measurement report, elucidating the actual underlying processes is beyond its scope. Nevertheless, in addition to the study of Frege et al. (2017) which we already mention, we have added a reference to the paper by Mauldin III et al. (1999), which also reports a connection between low RH and higher gas-phase sulfuric acid, and we now mention as well the work of Tsagkogeorgas et al. (2017), which would support the hypothesis of sulfuric acid evaporation from the condensed phase:

*"Similar observations were reported by Frege et al. (2017) at the high altitude station of Jungfraujoch (3454 m a.s.l., Switzerland) and Mauldin III et al. (1999) from airborne measurements over the Pacific Ocean. Based on the work of Tsagkogeorgas et al. (2017), the connection between increased sulfuric acid concentration and lower RH might be explained by the evaporation of sulfuric acid from the condensed phase, which they report as the main driver of sulfur particle shrinkage at low RH combining chamber experiments and model simulations."*

**Comment 13**: Page 20 Line 15. The reviewer was wondering how could MSA fragmentation in the mass spectrometer form SO5-? Could the authors add a reference to support?

We did not formulate this hypothesis with the knowledge of any results that might support it in the literature, but on the basis of our observations which highlight a strong similarity in the signal variations of these two compounds. We have slightly modified the wording of the sentence to make this clearer:

*"Finally, given the very high correlation observed between the signals of the MSA and SO₅⁻ groups, we cannot rule out the possibility that SO₅⁻ is a fragment of MSA in the mass spectra."*

**Comment 14**: Page 25 Line 10-11. The reviewer was wondering why would lower time spent over the ocean lead to higher wind speed along the air mass path. Could the authors clarify a bit and add a reference to support?

This explanation was only an assumption based on the correlations we observe between the signals of $SO_5^-$ and MSA and the time spent by the air mass over the ocean since it last flew over land. Since we have no strong argument or reference to support it, however, we have removed this part of the discussion from the revised version of the manuscript.

**Comment 15**: Figure 8. Considering April 16 early morning till 10 a.m. local time was in FT conditions with not too low radiation (Fig 4), the reviewer was curious whether this day was a NPF event day since SA and MSA levels were not low? If yes, is it possible to study e.g. few hours of FT nucleation/clustering process?

Based on the measurements performed with a Neutral Cluster and Air Ion Spectrometer (NAIS; Manninen et al., 2016), there is no clear evidence of NPF on that day (that would rather be classified as "Undefined" following the usual classifications; Fig. S7), which, as a reminder (see P13, L16-17), was characterized by unusual conditions likely linked to the arrival of tropical storm Fakir, which hit the island on the 19th. This information has been added to Sect. 4.1:

*"… an exception is observed on the morning of April 16, 2018 when FT conditions remain later at the station until 06:00 UTC (10:00 local time). One should note, however, that although daytime FT conditions could be sampled on that day, it unfortunately did not give the opportunity to study a case of MFT NPF as we could not identify clear evidence of NPF (this day was rather classified as "Undefined" based on the classification of Hirsikko et al. (2007); Fig. S7)."*

[Figure]

Fig. S7: Diurnal variation of a. negative ion and b. particle number size distribution measured with a NAIS on April 16th, 2018.

**Technical:**

Page 7 Line 7-8. Change to "the intensity of which vary…"

Done.

Page 7 Line 9. Change to "to a large extent".

Done.

Page 18 Line 2 and 14. Remove "the order of".

Ok, removed.

Page 18 Line 16. It only seems to be up to ~50% in Fig 7 on April 12. Please double check.

In fact the fraction shown in Fig. 7 for the sulfuric acid group does not exceed ~50% on April 12. The difference with the value reported in the text is explained by the fact that, as indicated on P15 L22-23, the results shown in Fig. 7 are averaged over 30 minutes to improve the clarity of the figure, while the values reported in the text were calculated on the 3-minutes averaged signals to retain the information associated with this higher temporal resolution.

Page 21 Line 15. Remove the duplicated "in".

Ok, removed.

Fig S4. It would be nice to swap the x axis from -12h on the left to 0h on the right side.

As suggested, the x-axis was swapped from -12h on the left to 0h on the right side.

References:

Ehn, M., Thornton, J. A., Kleist, E., Sipila, M., Junninen, H., Pullinen, I., Springer, M., Rubach, F., Tillmann, R., Lee, B., Lopez-Hilfiker, F., Andres, S., Acir, I. H., Rissanen, M., Joki nen, T., Schobesberger, S., Kangasluoma, J., Kontkanen, J., Nieminen, T., Kurten, T., Nielsen, L. B., Jorgensen, S., Kjaergaard, H. G., Canagaratna, M., Dal Maso, M., Berndt, T., Petaja, T., Wahner, A., Kerminen, V. M., Kulmala, M., Worsnop, D. R., Wildt, J., and Mentel, T. F.: A large source of low volatility secondary organic aerosol, Nature, 506, 476–480, https://doi.org/10.1038/nature13032, 2014.

Hirsikko, A., Bergman, T., Laakso, L., Dal Maso, M., Riipinen, I., Hõrrak, U., and Kulmala, M.: Identification and classification of the formation of intermediate ions measured in boreal forest, Atmos. Chem. Phys., 7, 201–210, https://doi.org/10.5194/acp-7-201-2007, 2007.

Leriche, M., Tulet, P., Deguillaume, L., Burnet, F., Colomb, A., Borbon, A., Jambert, C., Duflot, V., Houdier, S., Jaffrezo, J.-L., Vaïtilingom, M., Dominutti, P., Rocco, M., Mouchel-Vallon, C., El Gdachi, S., Brissy, M., Fathalli, M., Maury, N., Verreyken, B., Amelynck, C., Schoon, N., Gros, V., Pichon, J.-M., Ribeiro, M., Pique, E., Leclerc, E., Bourrianne, T., Roy, A., Moulin, E., Barrie, J., Metzger, J.-M., Péris, G., Guadagno, C., Bhugwant, C., Tibere, J.-M., Tournigand, A., Freney, E., Sellegri, K., Delort, A.-M., Amato, P., Joly, M., Baray, J.-L., Renard, P., Bianco, A., Réchou, A., and Payen, G.: Measurement Report: Bio-physicochemistry of tropical clouds at Maïdo (Réunion Island, Indian Ocean): overview of results from the BIO-MAÏDO campaign, EGUsphere [preprint], https://doi.org/10.5194/egusphere-2023-1362, 2023.

Manninen, H. E., Mirme, S., Mirme, A., Petäjä, T., and Kulmala, M.: How to reliably detect molecular clusters and nucleation mode particles with Neutral cluster and Air Ion Spectrometer (NAIS), Atmos. Meas. Tech., 9, 3577–3605, https://doi.org/10.5194/amt-9-3577-2016, 2016.

Mauldin III, R. L., Tanner, D. J., Heath, J. A., Huebert, B. J., and Eisele, F. L.: Observations of H2SO4 and MSA during PEM-Tropics-A, J. Geophys. Res.-Atmos., 104, 5801–5816, https://doi.org/10.1029/98JD02612, 1999.

Rocco, M., Colomb, A., Baray, J.-L., Amelynck, C., Verreyken, B., Borbon, A., Pichon, J.-M., Bouvier, L., Schoon, N., Gros, V., Sarda-Esteve, R., Tulet, P., Metzger, J.-M., Duflot, V., Guadagno, C., Peris, G., and

Brioude, J.: Analysis of Volatile Organic Compounds during the OCTAVE Campaign: Sources and Distributions of Formaldehyde on Reunion Island, Atmosphere, 11, 140, https://doi.org/10.3390/atmos11020140, 2020.

Tsagkogeorgas, G., Roldin, P., Duplissy, J., Rondo, L., Tröstl, J., Slowik, J. G., Ehrhart, S., Franchin, A., Kürten, A., Amorim, A., Bianchi, F., Kirkby, J., Petäjä, T., Baltensperger, U., Boy, M., Curtius, J., Flagan, R. C., Kulmala, M., Donahue, N. M., and Stratmann, F.: Evaporation of sulfate aerosols at low relative humidity, Atmos. Chem. Phys., 17, 8923–8938, https://doi.org/10.5194/acp-17-8923-2017, 2017.

Yan, C., Nie, W., Äijälä, M., Rissanen, M. P., Canagaratna, M. R., Massoli, P., Junninen, H., Jokinen, T., Sarnela, N., Häme, S. A. K., Schobesberger, S., Canonaco, F., Yao, L., Prévôt, A. S. H., Petäjä, T., Kulmala, M., Sipilä, M., Worsnop, D. R., and Ehn, M.: Source characterization of highly oxidized multifunctional compounds in a boreal forest environment using positive matrix factorization, Atmos. Chem. Phys., 16, 12715–12731, https://doi.org/10.5194/acp-16-12715-2016, 2016.

---

## Referee Report (RR1)

I sincerely appreciate that the authors have carefully considered the comments and concerns I had raised previously. Most of my comments have been addressed satisfactorily. I will elaborate on the two exceptions (Comments 1b and 3) below.
Otherwise, I have found the revised paper well structured and written, and following comprehensive and scientifically sound procedures for analyzing a novel ambient dataset. I believe re-addressing my remaining concerns will amount to only minor additional revisions, subject to which I gladly recommend the manuscript for publication.

**Regarding "Comment 1b":**

It may be worth clarifying unambiguously, which "spikes" I have been writing about: the spikes in the C4 amines time series, some of which I marked by orange circles I added below to the manuscript's Fig. 8b:

[Figure]

These spikes occur throughout the measurement period, both well within the white and green patches, representing BL and FT conditions respectively. Assuming we have all been talking about the same thing, the authors' response (and the manuscript at now P24 L15-16) contribute these spikes ("peaks") to mixing of BL and FT air "at the interface between the BL and the FT". My issues with that are:

- How to conclude that FT/BL mixing is happening in those instances? I would expect that to happen at the transitions between green and white patches, but the spikes occur throughout.

- If mixing was the cause of these spikes, what would be candidate mechanism? It would unlikely be a concentration differential between the different air masses, because that differential should at least to some degree become apparent once the BL grows to station elevation (or collapses beneath it). Could some evaporative process be responsible?

- Interestingly, as pointed out in the authors' response to "Comment 2", several other mass spectral peaks exhibit concurrent spikes (Fig. S14). Maybe I get too excited about those spikes now, but it is really unfortunate that those other peaks have remained unidentified, as their underlying compositions may reveal an explanation for what was going on.

With that in mind, I would like to ask again if the mass defect of those compounds could at least provide some clues regarding what kind of compositions are possible or can be excluded? They are mostly major peaks in the spectra. If m/z 198 and 261, for instance, are identifiable as $C_4H_{11}N.(HNO_3)_{1-2}.NO_3^-$, the mass defects of m/z 246, 260, etc. should at least be constrainable to a useful degree?
Alternatively, how certain are actually identifications like those for m/z 198 and 261?

So, to sum up my concerns here:
1) The connection of those spikes to BL/FT mixing, as currently put forward, requires further explanation.
2) If maintained, one or more mechanisms should be speculated on.
3) The involved mass defects (and/or uncertainties in peak identifications) should be at least discussed at some level.

**Regarding "Comment 3":**

First, I apologize that my sentence containing "ion clusters" was badly phrased (at the least I shouldn't have used the """""). It was not meant as a separate comment, but only to explain why I believe the use of "cluster" is questionable in some of this paper's context -- as the authors anyway correctly understood based on the first part of their response. Indeed, I have no issue with the original axis labeling in Fig. 7b.

As for the actual point, the authors argue in their response that molecules identified by CIMS are commonly referred to as "molecular clusters". Could the authors give evidence of that "accepted usage in the literature"? (I have honestly failed to notice, and a quick Google search did not help either. Indeed, Google only serves me counter-examples. But even if evidence was provided, I would probably still argue that we should strive for using accurate terminology.) Anyway, I ended up believing that we are still/again misunderstanding each other here.

Better to continue with the added text that now points out the usage of the "cluster" term in Section 4.2.1.
That is a good addition, but it does not settle the issue. The added text (correctly) defines the "cluster" term as referring here to clusters present in the atmosphere and clusters formed in the instrument. That is of course correct and perfectly fine, but in contradiction to the authors' response and different from the (incorrect) usage of "molecular clusters" in the title and abstract (L24). Also incorrect, I believe in at P4 L7 (Intro), P34 L11 (Conclusions), and Table 1, which hopefully covers it all.
To reiterate, I still find the use of "clusters" in the title ("[...] molecular clusters in the free troposphere [...]") inappropriate, as it is about >90% erroneous (Fig. 7b). And it also misleads in the abstract, especially given the beginning with reference to new particle formation, which involves actual molecular clusters. I further suggest fixing those couple of other instances too. (Unless, that is, I have really been missing key literature that uses the cluster terminology as referring, confusingly, to molecules.)

And I appreciate the authors pointing out more examples of actual atmospheric cluster observations. So, for the title, for instance, just adding "and potential precursor molecules" could be an acceptable solution.

---

## Author Response (AR2)

First of all, we would like to thank the Reviewer for once again taking the time to evaluate our work and help us to improve it. We have reconsidered below the two comments that required our attention and revised the manuscript accordingly.

Regarding Comment 1.b

We have followed the advice of the Reviewer to investigate further the peaks present on the C4 amines signal and the unidentified compounds showing the same peculiarity. In particular, we have plotted the mass defect (MD) plot corresponding to the data presented in Fig 6.a (shown below and also included in supplement, Fig. S11). We are very grateful to the Reviewer for his/her insistence on this point, since the complementary view provided by the MD plot and the additional analysis we carried out in this context enabled us to highlight a similarity in the behaviour of C4 amines and identified fluorinated compounds. This suggests that the signal of C4 amines is (at least to some extent) affected by a contamination, and we have therefore decided to exclude these compounds from the analysis proposed in section 4.2.2 (consequently, slight modifications have also been made to the abstract and conclusion).

[Figure]

Figure S11 Mass defect plot (corresponding to the data shown in Fig. 6.a) showing the identified species (marked by their chemical composition) as well as non-identified compounds associated to major peaks in the spectrum (marked by their mass, in bold). The size of the markers is related to the signal intensity (logarithmic scale).

The main changes made to the text (Sect. 4.2.1) in connection with this point are indicated below:

*P13 : "In order to offer a different perspective on the data, Fig. S11 presents in addition the mass defect plot (corresponding to the data presented in Fig. 6.a) highlighting the abovementioned species as well as the signals of the non-identified compounds associated to major peaks which are briefly discussed at the end of the section."*

*P18: "In contrast, while similar behaviour is observed by Brean et al. (2021) for C2 and C4 amines, there is no marked diurnal cycle in C4 amines signal at Maïdo (which constitute an even larger fraction of the identified signal, on average 28.6% and 27.2% in the FT and in the BL, respectively, Fig. 7.b); sporadic peaks, which are further discussed below, are observed instead."*

*P18-19: "In the absence of blanks, in particular, the risk of a contamination in the measurements cannot be excluded. In the case of C4 amines, it appears that the peaks mentioned above are in fact similar to*

*those present in the signal of the identified fluorinated compounds (Fig. S12), and more generally there is a strong correlation between the signal of C4 amines and that of $H(CF_2)_4COOH \cdot NO_3^-$ ($R^2 = 0.60$) and $H(CF_2)_5COOH \cdot NO_3^-$ ($R^2 = 0.70$). The suspicion of a contamination is therefore high for these compounds, which will not be investigated further. For C2 amines, in contrast, there is no such correlation with fluorinated compounds, and there is also no correlation with temperature (Fig. 10), which further rules out the possibility that these compounds are the result of a temperature modulated contamination by the instrument or the sampling line (volatilization-related)."*

*P20: "The other masses (UMR 246, 260, 310, 344 and 362 Th) have a different behaviour (Fig. S15.b), which is actually comparable to that observed for the peaks attributed to C4 amines (Figs. 8 and 9) and fluorinated compounds (Fig. S12), suggesting that these compounds could once again be the result of a contamination. In particular, as illustrated on the mass defect plot shown in Fig. S11, the peaks identified at UMR 260 and 310 Th have a mass difference corresponding to the mass of the $CF_2$ group, suggesting that these compounds may actually be fluorinated species."*

**Regarding Comment 3**

In order to avoid introducing any approximations in the nomenclature, we have followed the Reviewer's recommendations and proscribed the use of the expression "molecular clusters" or completed it where it was not the most relevant (including in the title). We have also removed the sentence added at the previous review stage (and mentioned by the Reviewer in this new comment), whose purpose was to warn of the use of the approximation that we have finally avoided.